

# Is global dimming and brightening in Japan limited to urban areas?

Katsumasa Tanaka[1,2], Atsumu Ohmura[2], Doris Folini[2], Martin Wild[2], Nozomu Ohkawara[3]

[1]Center for Global Environmental Research, National Institute for Environmental Studies (NIES), Tsukuba, Japan
[2]Institute for Atmospheric and Climate Science, Swiss Federal Institute of Technology (ETH), Zurich, Switzerland
[3]Global Environment and Marine Department, Japan Meteorological Agency (JMA), Tokyo, Japan

*Correspondence to*: Katsumasa Tanaka (tanaka.katsumasa@nies.go.jp)

**Abstract.** Observations worldwide indicate secular trends of all-sky surface solar radiation on decadal time scale, termed *global dimming and brightening*. Accordingly, the observed surface radiation in Japan generally shows a strong decline till the end of the 1980s and then a recovery toward around 2000. Because a substantial number of measurement stations are located
within or proximate to populated areas, one may speculate that the observed trends are strongly influenced by local air pollution and are thus not of large-scale significance. This hypothesis poses a serious question as to what regional extent the global dimming and brightening are significant: Are the global dimming and brightening truly global phenomena, or regional or even only local? Our study focused on 14 meteorological observatories that measured all-sky surface solar radiation, zenith transmittance, and maximum transmittance. On the basis of municipality population time series, historical land use maps,
recent satellite images, and actual site visits, we concluded that eight stations had been significantly influenced by urbanization, with the remaining six stations being left pristine. Between the urban and rural areas, no marked differences were identified in the temporal trends of the aforementioned meteorological parameters. Our finding suggests that global dimming and brightening in Japan occurred on a large scale, independently of urbanization.

## 1 Introduction

Efforts have been put into place for centuries around the world to understand the surface energy budget of the earth (Ohmura, 2014). One of the pillars of such activities to date is the Global Energy Balance Archive (GEBA) project (Ohmura and Lang, 1989), which established a database for the measurements of Surface Solar Radiation (SSR) and other parameters around the world (www.geba.ethz.ch). A major finding from the GEBA project was the *global dimming and brightening* phenomenon (Ohmura and Lang, 1989; Stanhill and Cohen, 2001; Wild et al., 2005), which referred originally to the secular trend of all-
sky SSR on the decadal time scale in Europe that had declined till around 1980s and then had been rising ever since (note that "global" refers to the sum of the direct and diffuse radiation rather than the geographical coverage). Secular trends under all-sky conditions have also been found elsewhere in the world, but the strength and the direction of the trend differ across regions (Wild et al., 2005; Ohmura, 2009; Wild, 2012).

It has been debated whether the global dimming and brightening is caused by aerosols and/or cloud effects (Stanhill
and Cohen, 2001; Wild, 2009). Although such effects are not entirely independent nor mutually exclusive because of aerosol-



cloud interactions (i.e. aerosol indirect effects), we discuss these two factors separately for the sake of discussion below. Water vapor also influences the all-sky SSR – however, the magnitude of its impact is substantially smaller than those from aerosols effects (Kvalevåg and Myhre, 2007; Ohmura, 2009; Wild, 2009). Now, dating back to early studies, (Ohmura and Lang, 1989) proposed cloud changes as the cause of global dimming, while (Stanhill and Moreshet, 1992; Liepert et al., 1994) attributed

aerosols to the radiation changes. Previous studies till now generally pointed to the dominant role of aerosols in bringing about the global dimming and brightening in China and Japan (Kaiser and Qian, 2002; Hayasaka et al., 2006; Qian et al., 2006; Norris and Wild, 2009; Kudo et al., 2012; Wang et al., 2012; Wang and Yang, 2014), in Europe (Norris and Wild, 2007; Ruckstuhl et al., 2008; Wang et al., 2012), and globally (Wild et al., 2005) based on a variety of approaches to estimating or inferring clear-sky (or cloud-free) SSR. (Streets et al., 2006) showed that the temporal trends of global-mean aerosol optical

depth (AOD) derived from emission inventories for sulfate dioxide ($SO_2$) and black carbon (BC) have resemblance with the global brightening and dimming phenomenon. (Ohmura, 2009) argued that the contributions from aerosols and clouds to the global dimming phase are roughly equal based on the transmittance data at several sites around the world. (Tsutsumi and Murakami, 2012) stresses the possible role of clouds, especially those arising from aerosol-cloud interactions, in shaping the global dimming and brightening in Japan. A fewer number of studies show that clouds effects are more dominant contributors

to the secular SSR changes (e.g. (Liepert, 2002) for the United States; (Liley, 2009) for New Zealand; (Russak, 2009) for Estonia; (Sanchez-Lorenzo and Wild, 2012) for Switzerland during the dimming period). The preceding discussion indicates that the answer to the question as to whether a secular SSR change is caused by aerosols or clouds is yet equivocal and varies across the regions as well as the periods. Our argument below proceeds with the premise that the global dimming and brightening in Japan is mainly caused by aerosols (including the effect from aerosol-cloud interactions) rather than clouds.

A remarkable feature in the observed all-sky SSR is the magnitude of its decadal change, which is as large as about 15 W/m$^2$ in Europe and amounts to up to 25 W/m$^2$ in Japan during 1960-1990 (Skeie et al., 2011). It is a subject of debate that the observed decadal SSR trends are generally stronger than those simulated by a number of models, although such trends in several different regions of the world are qualitatively well reproduced by some of the models (Ruckstuhl and Norris, 2009; Dwyer et al., 2010; Folini and Wild, 2011; Skeie et al., 2011; Wild and Schmucki, 2011; Allen et al., 2013). From both model

and observation perspectives, hypotheses have been put forward to explain the discrepancy between the observed and simulated SSR trends as summarized below:

(i)   From the model perspective, four conceivable reasons are: a) the resolutions of the models used in these studies are too coarse to capture the effects of local air pollution on SSR (Skeie et al., 2011), b) the models do not describe aerosol and cloud effects and their interactions sufficiently (Skeie et al., 2011; Allen et al., 2013), c) historical aerosol emission

inventories are incorrect (Ruckstuhl and Norris, 2009; Folini and Wild, 2011; Allen et al., 2013), and d) inaccurate variations in the aerosol fields used in the models (Wild and Schmucki, 2011).

(ii)   From the observational perspective, the discrepancy may be explained by the fact that a large number of measurement stations are located within or close to populated areas. It can be hypothesized that the observed all-sky SSR decadal shifts may not have universally happened and can be locally-confined urban effects from atmospheric pollution with the direct



aerosol effects being predominant (Kvalevåg and Myhre, 2007). Other issues that have been raised include the measurement data quality in some regions (Tang et al., 2011; You et al., 2013; Wang, 2014) and the spatial representativeness of stations (Hakuba et al., 2013).

The answer is yet inconclusive and the question remains debated (Wild, 2009). While statistical studies estimating
the level of human influence on SSR data based on demographic data claimed that the global dimming and brightening was limited to urban areas (Alpert et al., 2005; Alpert and Kishcha, 2008), several studies (Ramanathan et al., 2007b; Stanhill and Cohen, 2009; Wang et al., 2014; Imamovic et al., 2016) questioned the validity of their underlying assumption that the population was an effective proxy for human impacts on SSR. Furthermore, even if the geographical distribution of the stations is skewed strongly toward polluted locations, air pollution from urban areas would not be confined to peripherals of urban
areas due to the long-range transport as demonstrated by a number of observational and model-/satellite-based studies (e.g. (Akimoto, 2003; Pinker et al., 2005; Lawrence et al., 2007; McConnell et al., 2007; Ramanathan et al., 2007a; Yu et al., 2008; Folini et al., 2009; Monks et al., 2009; Yu et al., 2012; Yu et al., 2013; Lin et al., 2014)). Furthermore, anthropogenic emissions from sources such as coal combustion and biomass burning are known to influence even the stratospheric aerosol levels (Hofmann et al., 2009; Solomon et al., 2011).

This study aims to contribute to the debate surrounding the hypothesis from the observational side. This study is a first attempt to clarify exclusively whether the global dimming and brightening in Japan has been observed only in urban areas. To do so, we look into the data for SSR and maximum/zenith transmittance and inspect if there are any significant differences between their temporal trends over urban areas and those over rural areas.

Our approach can be characterized in comparison to other studies as below:

(i)  Earlier studies (e.g. (Imamovic et al., 2016)) deal with a large number of observatories worldwide (on the order of hundreds) that provide all-sky SSR data. On the other hand, this study focuses on a small set of stations in Japan (only 14 stations) (Figure 1) that yield not only SSR data but also transmittance data. This brings our approach an advantage to separate aerosol and cloud effects. Locations of the 14 stations analyzed in our study cover most of Japan homogeneously.

(ii)  While a few earlier studies (e.g. (Alpert et al., 2005; Alpert and Kishcha, 2008)) rely primarily on population data to infer
the influence of urbanization on SSR measurements, we collect and utilize as much information as possible. Those include historical land use maps, recent satellite images, and current photographs taken during actual site visits. Such geographical information has not been brought up to the debate associated with the causes of the global dimming and brightening. Furthermore, we look to station-specific historical information (e.g. site relocations and measurement instruments) compiled from archived documents. Such pieces of information, particularly those during the previous century, are not
digitalized and practically inaccessible without dedicated efforts.

(iii)  For the specific case of Japan, there are two related studies (Norris and Wild, 2009; Kudo et al., 2012). Both studies derived aerosol and cloud effects in the Japanese SSR data, but none of them addressed their impacts from urbanization, which we focus on. (Kudo et al., 2012) employs a method to estimate AOD and single scattering albedo from global and direct irradiance data measured at the same 14 stations. (Norris and Wild, 2009) uses two different cloud datasets to



estimate aerosol contributions to measured all-sky SSR levels. In contrast, our analysis directly makes use of the data for global SSR and zenith/maximum transmittance data.

The paper is organized as follows. In Section 2, we describe environmental changes surrounding the 14 stations based on available information and then classify the stations according to the inferred pollution levels. Such discussions on the station

surroundings are reflected in Section 3, which discusses the temporal changes in SSR and transmittance data in the urban and rural areas as well as at each individual station. Concluding remarks are provided in Section 4.

## 2 States of urbanization surrounding the observatories

### 2.1 Data related to urbanization

To investigate the possible effect of local air pollution on the observed SSR and transmittance, we separate the 14 Japanese

stations into two groups: *polluted* and *pristine* stations (Figure 1). The immediate surroundings of the former group are considered to have already been affected by urban development, while those of the latter are regarded as having being hardly influenced by urban development till present. More technically, a pristine station must satisfy both of the two "pristine conditions": i) it is currently located away from cities and ii) it has not seen any significant urban development in the peripheral region.

Our site classification is carried out on the basis of various sources of evidence at the municipality level. Indices that can be used as a proxy for urbanization are, however, generally scarce at the municipality level. There are no time-series that directly capture the air pollution level on the municipality scale for a sufficiently long period of time for this study. Relevant data are more abundant at the prefectural level, but only those at the municipality level are meaningful for our analysis focusing on local pollution. We therefore resort to an educated guess based on available information at hand. To put it another way,

because no existing single index can reliably and universally represent urbanization, we look to multiple lines of evidence. The evidences we bring include temporal changes in municipality population counts and densities since 1920 based on census (Figures 10 and 11), vehicle $CO_2$ emissions in 2003 (http://www.colgei.org/CO2/), the number of registered vehicles per capita in 2011 (Sangyochiiki_Kenkyusho, 2012) (Figure 2), historical land use maps during three selected periods (1955-1960; 1975-1980; 1995-2000) (Figures 5 and 6), recent satellite images taken during 2012-2015 (Figure 3), and current photographs taken

during the site visits in 2015 and 2016 (Figure 9). While there are several different sources of evidences used to evaluate the first pristine condition discussed above, the second pristine condition can be investigated only by changes in population counts and densities, historical land use maps, and recent satellite images.

Population data are the most accessible and only available long time-series on such a fine scale (note that municipality population data before 1970s are not systematically available in a digitalized format). There are, however, known limitations

when they are interpreted as proxies for urbanization. The use of population alone as a proxy to infer the air pollution level is under debate (Alpert et al., 2005; Ramanathan et al., 2007b; Alpert and Kishcha, 2008; Stanhill and Cohen, 2009; Wang et al., 2014; Imamovic et al., 2016). Related are issues associated with the use of population-based proxies to infer $CO_2$ emissions





(for an overview, see (Andres et al., 2012)), even though $CO_2$ emissions are not necessarily correlated with $SO_2$ emissions on the relevant spatial and time scales (Grossman and Krueger, 1995; Holtz-Eakin and Selden, 1995; Dinda, 2004). Population data may serve as a proxy for $CO_2$ emissions from residential and commercial sectors but work poorly for emissions from power and transport sectors. With respect to power sectors, population fails to represent $CO_2$ emissions when coal is combusted

in a remote area to support the electricity demands in a distant urban area. This type of issues becomes important when one deals with emission data on a fine spatial scale. Concerning transport sectors, transport emissions per capita are known to decline above a certain population density threshold (Gately et al., 2015). In our analysis, the population time-series are therefore complemented by the series of historical land use maps, recent satellite images, and current photographs, which provide additional insights into how the site surroundings have been changed during the past period of interest.

We employ about 50 maps containing the locations of the 14 observatories under study (including previous site locations if relocated) (Figure 8). Maps were selected to cover roughly the three different windows of period: i) 1955-1960, ii) 1975-1980, and iii) 1995-2000. Note, however, that some of the maps we use are slightly outside of these windows due to limited map availability. The choice of these time windows reflects our general observation that the urban development in many parts of Japan continued throughout the previous century, but in some areas it has slowed down after around 1980s,

which is parallel to the period when clean air policies entered into force across the country. One can thus compare the differences in the maps between periods i) and ii) with those between periods ii) and iii) in order to distill insights into how urbanization proceeded across the presumed turning point in peripherals of the sites. We use only the maps in the scale of 1:50000 since maps in such a scale were more frequently updated during the past century than those in a finer scale of 1:25000. To offer different levels of perspective, each map is shown in two spatial scales: one with a scale bar length of 1km (Figure 5)

and the other with a scale bar length of 4km (Figure 6). The maps we use were published from the Geospatial Information Authority of Japan (GSI) (or previous associated organizations for the maps roughly before 1960). Because maps of our interest are not available in a digital format, we scanned the paper maps at first. We then georeferenced the digitalized maps by using the software package QGIS and apply Google Earth Pro to present them geographically in relation to the current and previous site locations. These maps are supplemented by recent satellite images retrieved during years 2012-2015, which were also

obtained from Google Earth Pro. To provide a wider perspective, the satellite images are presented on three spatial scales (i.e. scale bar lengths are set at 200m, 1km, and 30km). It should be noted that the locations of some previous sites are only approximately shown on the maps and images because in some instances, the exact locations cannot be easily traced back from the addresses in old formats. The address format was altered over the long period of time, for instance, due to municipality mergers and dissolutions, which we discuss below.

Another issue associated with municipality data is frequent changes in jurisdictional boundaries due to municipality mergers and dissolutions. Municipal mergers alone can mimic a population increase of the afflicted cities and towns in statistics. Over a long period of time, the number of municipalities in Japan has been reduced from 71,314 (December 1888) to 1,713 (December 2015). The last two decades saw a halving of the number of municipalities because of enhanced merger activities around 2005. We account for such boundary changes in compiling the time series of population counts and densities (Figures





10 and 11) and highlight the biases that can be caused by boundary changes (Figure 10). The frequent boundary changes place limitations on the usefulness of other time-series at the municipality level even when they exist. For instance, data on the gross domestic production at the municipality level for the past several decades, which are available at some municipalities investigated in our analysis, are fraught with this problem.

Our educated guess about the possible urbanization impact on the SSR measurements draws on the experiences of A. Ohmura, a co-author who has been carrying out a long-term engagement in radiation measurements and has paid visits over the past several decades to a number of observatories around the world (Ohmura, 2009, 2014), as well as the recent activities by K. Tanaka, the first author who visited seven sites out of the 14 under study (Akita, Matsumoto, Shimizu, Shionomisaki, Tateno, Yonago, and Wajima). To convey some impression of the present environments surrounding the sites, we present three
photographs taken at and around each of the seven sites capturing the measurement fields and their immediate surroundings within a radius of a few hundred meters (Figure 9). Changes in the immediate surroundings of the stations are crucial information for this study and available only by visiting the sites (e.g. see the discussion on Wajima in Section 2.2). We argue that the selection of meteorological stations appropriate for a long time-series analysis requires a field investigation focusing on changes in the stations and their environments. For instance, changes in instrumentation and instrument locations are best
available in the archives at meteorological headquarters as well as at sites. One should be cautious about studying only published statistics without visiting sites, as relevant changes are often unnoticeable in statistics.

## 2.2 Classification of the polluted and pristine stations

With the preceding considerations, we arrived at six stations (i.e. Ishigakijima, Miyako, Nemuro, Shimizu, Shionomisaki, and Wajima) that had not been flagged for pollution. They are thus qualified for the pristine group, while the remaining eight
stations (i.e. Akita, Fukuoka, Kagoshima, Matsumoto, Naha, Sapporo, Tateno, and Yonago) fall into the polluted group. The rationales for such site classifications are described below. For the sake of discussion, stations are described according crudely to the order of current development. It should be noted that there remain, however, counter-evidences that favor the Ishigakijima and Wajima sites in the polluted camp. To address such counter-evidences, we perform sensitivity analyses by changing the group compositions with respect to these sites.
• *Stations in large cities (Sapporo and Fukuoka)*: It is virtually certain that Sapporo and Fukuoka observatories have been affected by urbanization. These stations are located within major cities in Japan, whose current populations are above 1,000,000 (Figure 2) and whose built-in areas are extensive in space (Figure 1), which lead to a violation of the first pristine condition introduced in Section 2.1. Both of the stations have been in operation at their respective current locations without any relocations since 1939 (Figure 2). The rapid expansion of the Sapporo city has come during the first half of
the 20th century and has slowed down around 1980 (Figures 5 and 6), which is also evident in the trends of population count and density (Figures 10 and 11). The development of the Fukuoka city has taken off later during the last century (Figure 6) accompanied by the steady population growth till present (Figure 10). Such city growths were considered as violations of the second pristine condition.





- *Station in a medium-size city (Kagoshima)*: The station is located in a medium-size city (current population: about 600,000) and characterized by the large population density and vehicle $CO_2$ emissions (Figure 2). The city occupies a large area (Figure 1). Visual inspection of the land use changes and the population trends (Figures 5, 6, 10, and 11) suggested that the city growth is continuing till present but with a declining rate since 1980s. The site has been moved in 1994 with a distance of a few kilometers (Figure 2).

- *Stations in small cities (Akita, Matsumoto, Naha, Tateno, and Yonago)*: These stations are found in smaller cities (current population: about 150,000-300,000) and discussed one by one in the alphabetical order in the following:

  *Akita*: The previous (1926-1989) and current (1989-present) sites are located within a business district and just a few hundred meters away from the prefecture government building. Highways pass around the current site and there are large above-ground parking lots nearby (Figures 1, 3, and 7). Both the previous and current sites have been gradually swallowed by the surrounding urban development (Figures 5 and 6).

  *Matsumoto*: Among other sites under study, this is the only site located at a relatively high altitude (610m). The current site has been in operation since 1935 (Figure 2). The series of historical land use maps (Figures 5 and 6) showed that the Matsumoto city had gradually expanded to the area encompassing the site. The immediate surrounding of the site is currently residential, punctuated by a few tall apartments (Figure 9). Roads with relatively heavy traffic passing by the Shinshu university campus run about a few hundred meters away (Figure 5).

  *Naha*: Although the Naha city is relatively small in terms of the population count, the city currently has the highest population density among the 14 municipalities in spite of its location away from the Japanese mainland (Figure 2). The population count and density of Naha grew rapidly during 1950-1975 (Figures 10 and 11). The observatory was relocated just once during the study period but moved several times previously (Figure 2 and Figure 6) as discussed earlier as an extreme case of station relocations.

  *Tateno*: The site is located in the Tsukuba city, a city newly developed for research and development whose major transport mode is vehicle as indicated by the relatively large number of vehicles per capita (Figure 2). The site has been invaded by urban development that took off around 1960s although it is not as dense as in the other major urban areas. Currently there is a major road passing relatively close to the site (Figures 1, 5, and 9). The site location has been stable without any substantial relocation since its inauguration in 1920 (Figure 2).

  *Yonago*: The Yonago city is located in a currently remote area of Japan. The city area has spread over the site gradually during the past century (Figures 5 and 6). The population density in Yonago has been as high as that in Kagoshima since around 1980 (Figures 2 and 11). The site has not been moved for a long period of time (Figure 2). The immediate surrounding of the site is residential, but there are major roads about a few hundred meters away from the site (Figures 1, 5, and 9).

- *Stations in rural areas (Ishigakijima, Miyako, Nemuro, Shimizu, Shionomisaki, and Wajima)*: No strong indication was found that air pollution could have affected these six sites. Remarks for each station are given in the alphabetical order below:





*Ishigakijima*: The station is regarded as a pristine site based primarily on the expert judgment of A. Ohmura. In spite of its distant location from the Japanese mainland, visual inspection of the land use maps (Figure 6), however, indicated a tantalizing sign of small-scale urbanization that has undergone around the site. While the projection of population count in Ishigakijima is more comparable with those associated with other pristine stations (Figure 10), the projection of population density in Ishigakijima is close to that in Matsumoto, a polluted station (Figure 11). The category of this station is thus subject to a sensitivity analysis.

*Miyako*: The population density of Miyako has been the lowest for the past 50 years of the sites considered here (Figure 11) and the population in Miyako has been decreasing since 1960s (Figure 10). The area surrounding Miyako has been unaffected by any city expansion. However, the latest satellite images of Miyako (Figure 3) do not accurately portray the stable pristine surroundings that have persisted (Figures 5 and 6) because the surrounding of the site has been heavily affected by the 2011 Tohoku earthquake and tsunami (Figure 4). The series of historical maps indicated that there had been no major urban activities in this area prior to the disaster (Figures 5 and 6).

*Nemuro*: The site location has been unchanged since 1886 (Figure 2). It is located in a remote area at the southeastern edge of Hokkaido Island with no major changes in the surrounding for a long period of time (Figures 1, 5, and 6). The population count and density have not been significantly changed over the course of 100 years (Figures 10 and 11).

*Shimizu*: The site finds itself in a region known for the untouched nature and detached from industrial activities except for tourism and fishing. The station is located in one of the least accessible areas in Japan – the Shimizu area is the southernmost point of Shikoku Island with limited road connections, reflecting its geographical and topographical characteristics surrounding the area (Figure 3). The area extent and the population of the nearby settlement have been stable during the study period (Figures 6, 10, and 11). The site location has been intact since its foundation in 1940 (Figure 2) and is surrounded by the ocean, residences, and graveyards (Figures 3 and 9).

*Shionomisaki*: This site is located in a poorly accessible area confined between the Kii mountain ranges and the Pacific. The population count and density in the Kushimoto town, in which the site is located, have not been substantially changed over the past 100 years with a small declining trend since 1950s (Figures 10 and 11). The land use around the site has been mainly residential (Figure 5). The extent of the built-in area around the site has been almost the same during the study period (Figure 6). The observatory was inaugurated in 1912 and relocated in 2009 with a distance of approximately 350m (Figure 2). The new site is surrounded by graveyards and residences (Figures 3 and 9).

*Wajima*: The Wajima site is located in a small traditional town culturally known for woodworking and lacquer. The town has not undergone any significant urbanization till present (Figures 5 and 6). Since 1929 the site has not experienced any relocation with a significant distance (Figure 2). The population count and density in Wajima are comparable to those in other pristine areas (Figures 10 and 11). During the site visits by K. Tanaka, however, wood-burning factories were observed just a few hundred meters away from the site today. He witnessed a smokestack in operation during his visits on 20 November 2015 and 28 March 2016 (Figure 9). Records suggest that the factories



have been in operation for the past several decades, leading to a speculation that the smokestack might have affected the radiation measurements over a long period of time. Similar to the treatment for Ishigakijima, this site classification is put to the sensitivity analysis.

With the foregoing station separations based on our best knowledge, we proceed to the SSR and transmittance data analysis in the next section.

## 3 SSR and transmittance at the observatories

### 3.1 History of the SSR measurements in Japan

We begin with a brief historical background of the SSR measurements in Japan. SSR measurements have started in 1940 in Sapporo, followed by Nemuro in 1941 and Miyako in 1942 according to Kishokanpo (in Japanese), the activity records of meteorological stations in Japan disclosed at the Japan Meteorological Agency (JMA). Earliest observations at most of the other sites started in 1950s. Robitzsch-type pyranometers were used for SSR measurements during this period. In the International Geophysical Year (IGY) (July 1957 – December 1958), more widespread SSR measurements have been initiated globally. It is not until 1961 when SSR data became officially available at the JMA website today. During the 60s, two different types of instrument (i.e. Eppley- and Robitzsch-type pyranometer) (Garg and Garg, 1993) were used in parallel (Figure 2). Both types of instrument have been replaced with Moll-Gorszynski thermopile pyranometers at all sites in early 1970s, which is when instrumentally harmonized measurements have started at the 14 observatories (for an overview of pyranometers and other radiation measurement instruments, see (Emeis, 2010)). The period that follows saw several station relocations including those for Akita in 1989, Kagoshima in 1994, Miyako in 1990, Naha in 1987, and Shionomisaki in 2009 (Figure 2). Rather frequent station relocations are not unusual, with the Naha observatory that has been relocated seven times since its inauguration (Figure 6) being the most extreme case. Since 2007-2010 the radiation measurement campaign has been reduced in scale. This move is partly due to the budget cut, but the reduction in SSR sites is not unrelated to the recent more widespread use of sunshine duration sites to study the global dimming and brightening (Sanchez-Romero et al., 2014). Several observatories including Matsumoto, Miyako, Nemuro, Shimizu, Shionomisaki, Wajima, and Yonago (Figure 2) had been thus automated. However, automated radiation measurement systems have not been proved reliable over a long run. Currently, radiation measurements are not possible without staffs in residence. It is recommended that "pyranometers in continuous operation should be inspected at least once a day and perhaps more frequently, for example when meteorological observations are being made" (World_Meteorological_Organization, 2014).

### 3.2 Data used in our analysis

SSR data used in our study have been collected under the auspices of JMA (http://www.data.jma.go.jp/obd/stats/etrn/index.php), one of the most concerted and systematic efforts to archive data dating back to as early as 1870s (e.g. surface temperature and precipitation). The JMA dataset covers more than 100 observatories in





Japan and comprises various parameters. We deal with SSR data from 1961 onward, when SSR data are officially available. There are other data in JMA that can be applicable to our study, but our analysis focuses on the confined set of data that we are most familiar with. Note that we do not apply latitudinal corrections for the annual-mean sunshine angle and the effective atmosphere thickness because we inspect the SSR anomalies relative to the 1976-1995 levels, which are unaffected by

latitudinal effects.

In addition, we use zenith and maximum transmittance data (calculated from direct solar measurements) to support the analysis to inspect the effect of aerosols on SSR changes. Because the majority of studies indicates that the global dimming and brightening in Japan was predominantly caused by aerosols (Kaiser and Qian, 2002; Qian et al., 2006; Norris and Wild, 2009; Kudo et al., 2012), transmittance data, which are more directly related to aerosols because transmission measurements

are taken only under clear sky conditions, can be insightful. Data for these quantities are available since as early as 1930s. Mean estimates of the measurements at 9am, 12pm, and 3pm are used in our analysis. Both of the transmittance quantities have been measured by human eyes and are thus subject to human-induced bias. The presence of cirrus clouds, which can be missed out without a well-trained observer, can influence the measurements of zenith transmittance. While the measurements of maximum transmittance are free from this problem by definition and thus believed to have a higher fidelity, this parameter

suffers from a representativeness issue since it captures only the extreme in each measurement period. Given the advantages and disadvantages, we look into both zenith and maximum transmittance data in our analysis.

### 3.3 SSR trends at the polluted and pristine stations

All-sky SSR anomaly data averaged over all the stations indicated a dimming till late 1980s followed by a brightening (Figure 12), which is consistent with the findings in previous studies (Ohmura, 2009; Skeie et al., 2011; Kudo et al., 2012). Their

trends averaged over the polluted and pristine sites, which are more important to our analysis, showed no discernible differences. This suggests that local air pollution did not play a role in causing the global dimming and brightening phenomenon in Japan. The same indication was obtained from a few sensitivity cases using different station group compositions (Figure 15). It is also worth noting the gap in 1971 in SSR data, before which SSR anomalies at individual stations show higher variabilities and larger amplitudes of changes. This is mainly because of the less accurate instruments

used in earlier days, although this might also be related to the parallel use of two different types of instrument during the 60s (Figure 2).

### 3.4 Transmittance trends at the polluted and pristine stations

Further evidences to support the preceding conclusion (i.e. secular SSR changes persisting commonly at polluted and pristine sites) were found in the zenith and maximum transmittance. The zenith and maximum transmittance trends between the

polluted and pristine stations followed a similar path and did not diverge over time (Figure 13). Such an indication is not altered in the sensitivity cases using slightly different station groups (Figures 16 and 17). This suggests that air pollution does



not just locally control the transmittance. In other words, the indication here contradicts with the claim that air pollution owing to the continuing urbanization had affected only urban areas.

Measurements from single stations are not in conflict with the abovementioned conclusion although they are subject to larger uncertainties. The zenith and maximum transmittance in Sapporo exhibited steadily upward trends since around 1970 (Figure 14i). If the change in transmittance were driven by local air pollution, downward trends would instead be anticipated from the urbanization in Sapporo that has been continuing till now (Figures 5, 6, 10, and 11). A contrasting example is the zenith and maximum transmittance levels in Shimizu, both of which have been declining since 1950s (Figure 14j). The Shimizu station is known for its persistently clean surrounding since its inauguration, providing no reason locally to support the continuously declining transmittance trends. These findings reinforce the argument that temporal changes in transmittance are not a product of locally-confined air pollution.

## 3.5 Discussions

The foregoing data analysis generally indicated that the changes in SSR as well as transmittance are not convincingly explained by local air pollution. Such a conclusion is in line with burgeoning literature on long-range transboundary air pollution (e.g. (Ramanathan et al., 2007a)) and modeling studies indicating long-range transport in the order of hundred kilometers (e.g. (Folini et al., 2009)). Moreover, our general observation is that a majority of studies investigating SSR trends in different regions of the world converge to similar conclusions. For example, the findings of global dimming and brightening in the Arctic (Stanhill and Callaghan, 1995; Ohmura, 2006) and Antarctica (Stanhill and Cohen, 1997), areas virtually without any human interference, are additional proofs that these variations are not limited to urban regions.

The transmission measurements shown in the present study support the influence of aerosol, as they represent clear sky conditions and show dimming and brightening tendencies which should not be caused by clouds. This finding supports our premise that the global dimming and brightening in Japan is related more to aerosols than clouds as also indicated by other studies (Kaiser and Qian, 2002; Qian et al., 2006; Norris and Wild, 2009; Kudo et al., 2012). Because the transmittance is not predominantly shaped by local air pollution, the global dimming and brightening is not caused by local air pollution, but by large scale changes in background aerosols.

Now, a further question arises as to what has determined the overall trend of Japanese transmittance time series. First, the temporal changes in the zenith and maximum transmittance averaged over all the stations are punctuated by the two globally significant discrete events: one related to the El Chichón eruption (Mexico) in 1982 and the other due to the Pinatubo eruption (Philippines) in 1991, as also apparent in the transmittance records at Mauna Loa, Hawaii (Dutton and Bodhaine, 2001). The underlying trend of the all-station transmittance data is generally downward up until 1970s (Figure 13). We speculate that such a downward trend stems from the urbanization that started earlier (Figures 7, 10 and 11), bringing about air pollution, which was not confined locally but rather spread out to larger areas through long-range transport. Then, the transmission trend levels off since 1980s despite the continuing urbanization. We hypothesize that two additional competing factors have entered around this period: i) clean air policies that have started to be implemented in Japan also since 1980s, which contributed to reversing





the downward trend, and ii) long-range transport of the air pollution resulting from the rapid development of China started around the same period, which strengthened the downward trend. The abovementioned argument was inferred from the results of this study, but further research is required to verify the supposition.

## 4 Concluding remarks

We demonstrated multiple lines of evidence suggesting that the global dimming and brightening phenomenon in Japan was not restricted to urban areas and not primarily driven by local air pollution. The evidences include the similar temporal trends of the SSR anomaly and zenith/maximum transmittance data observed at the sets of polluted and pristine stations. Our station classification is based on an extensive collection of data and maps as well as actual station visits. The conclusion is robust against a few sensitivity cases using different station group compositions. The dimming and brightening in Japan is therefore a large-scale phenomenon, independently of local air pollution.

Due to the lack of data to separate aerosol and cloud effects during the historical period, future studies would also need to make use of various types of proxies to understand historical SSR changes. Our approach combining population data, historical land use maps, satellite images, and site visit experiences to infer the pollution level can be applied to regions and countries where such pieces of information are publically available in sufficient amount and also where stations are easily accessible. Furthermore, studies using high-resolution models with a capacity to address more realistic aerosol and cloud representations including cloud formation over urban area through physical features (Shepherd, 2005; Seto and Shepherd, 2009; Tao et al., 2012) might provide a new perspective. Our scientific understanding for this issue would be advanced further when more studies using contrasting approaches are performed and compared to each other.

*Acknowledgements.* K. Tanaka is partially supported by the Marie Curie Intra-European Fellowship within the 7th European Community Framework Programme (Proposal N°255568 under FP7-PEOPLE-2009-IEF). We thank Adel Imamovic (Swiss Federal Institute of Technology (ETH), Zurich) for contributing to the discussion related to this study. We are grateful for the introduction to measurement activities at Aerological Observatory in Tsukuba provided by Matsumi Takano (Aerological Observatory of Japanese Meteorological Agency). We thank Daisuke Murakami (National Institute for Environmental Studies, Tsukuba) for his technical supports to deal with GIS software packages and georeference the maps. Technical assistance provided by Kazuya Hirota (Atmosphere and Ocean Research Institute, University of Tokyo) to georeference a part of the maps used in this study is highly appreciated.

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



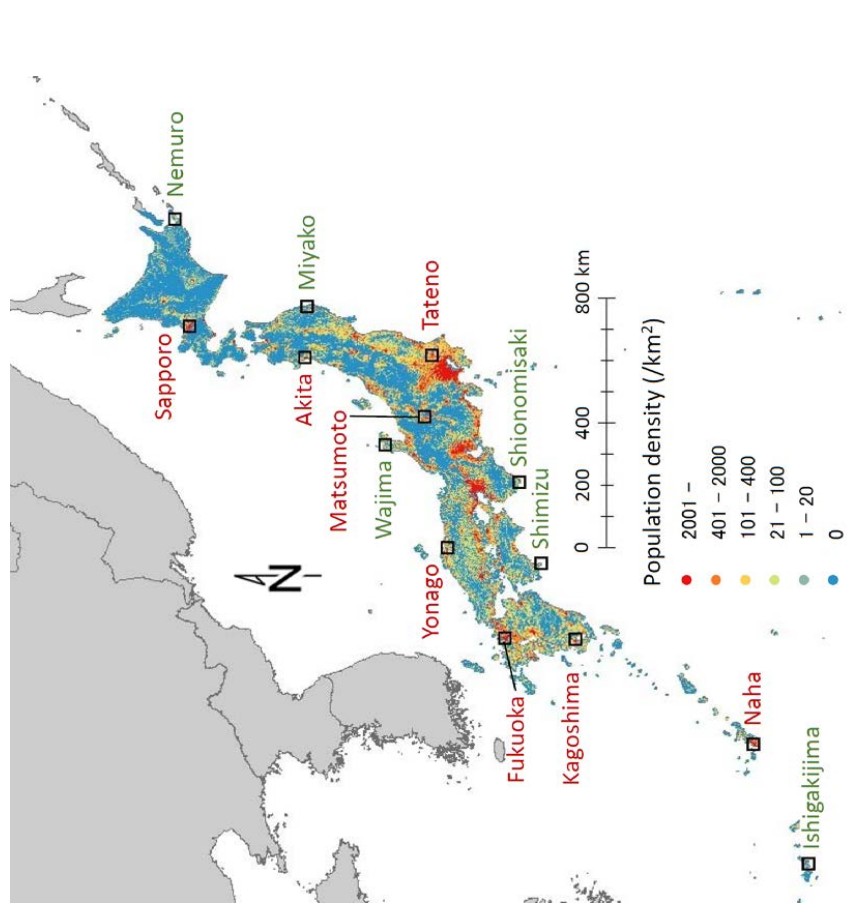

**Figure 1. Locations and classifications of the 14 Japanese stations used in our analysis. The names of the polluted and pristine stations are indicated in red and green, respectively. The underlying population density distribution is based on the 1km grid data for the 2005 population census of Japan (http://e-stat.go.jp/SG2/eStatGIS/page/download.html) and shown to indicate current urban areas.**



Atmospheric Chemistry and Physics — Discussions — Open Access — EGU



| Name / Official name in Japanese | Jurisdiction / Full address in Japanese | Latitude / Longitude | Altitude (m) | Year of foundation / (Year(s) of relocation) | Radiation measurement method during 1960s | Current staff residence | Municipality area (km²) | Population count and density (/km²) | Vehicle $CO_2$ emissions (t$CO_2$/year) / Vehicles per capita |
|---|---|---|---|---|---|---|---|---|---|
| Akita 秋田地方気象台 | Akita-city, Akita-pref. 秋田県秋田市山王 7-1-4 | 39°43.0'N 140°05.9'E | 6.3 | 1882 (1886, 1896, 1926[2], 1891[1]) | Robitzsch | Yes | 906.09 | 323,600 / 357 | 672,540 / 0.53 |
| Fukuoka 福岡管区気象台 | Fukuoka-city, Fukuoka-pref. 福岡県福岡市中央区大濠 1-2-36 | 33°34.9'N 130°22.5'E | 2.5 | 1889 (1907[2], 1939[1]) | Eppley | Yes | 343.38 | 1,463,743 / 4,263 | 1,996,277 / 0.37 |
| Kagoshima 鹿児島地方気象台 | Kagoshima-city, Kagoshima-pref. 鹿児島県鹿児島市東郡元町 4-1 | 31°33.3'N 130°32.8'E | 3.9 | 1883 (1897, 1915[2], 1994[1]) | Robitzsch | Yes | 547.57 | 605,846 / 1,106 | 1,160,781 / 0.49 |
| Matsumoto 松本特別地域気象観測所 | Matsumoto-city, Nagano-pref. 長野県松本市沢村 1-7-13 | 36°14.8'N 137°58.2'E | 610 | 1898[2] (1935[1]) | Robitzsch | No (2007-) | 978.47 | 243,037 / 248 | 493,205 / 0.58 |
| Naha 沖縄気象台 | Naha-city, Okinawa-pref. 沖縄県那覇市樋川 1-15-15 | 26°12.4'N 127°41.2'E | 28.1 | 1890[7] (1925[6], 1927[5], 1950[4], 1951[3], 1953[2], 1987[1]) | Robitzsch | Yes | 39.57 | 315,765 / 7,980 | 320,972 / 0.38 |
| Sapporo 札幌管区気象台 | Sapporo-city, Hokkaido-pref. 北海道札幌市中央区北 2 条西 18-2 | 43°03.6'N 141°19.7'E | 17.4 | 1876 (1878[3], 1890[2], 1939[1]) | Eppley | Yes | 1121.26 | 1,913,545 / 1,707 | 2,961,534 / 0.41 |
| Tateno 高層気象台 | Tsukuba-city, Ibaraki-pref. 茨城県つくば市長峰 1-2 | 36°03.4'N 140°07.5'E | 25.2 | 1920[1] (none) | Eppley | Yes | 283.72 | 214,590 / 756 | 327,686 / 0.57 |
| Yonago 米子特別地域気象観測所 | Yonago-city, Tottori-pref. 鳥取県米子市博労町 4-309-1 | 35°26.0'N 133°20.3'E | 6.5 | 1939[1] (none) | Robitzsch | No (2008-) | 132.42 | 148,271 / 1,120 | 314,391 / 0.56 |
| Ishigakijima 石垣島地方気象台 | Ishigaki-city, Okinawa-pref. 沖縄県石垣市字登野城 428 | 24°20.2'N 124°09.8'E | 5.7 | 1896 (1897[1]) | Robitzsch | Yes | 229.27 | 46,922 / 205 | 41,212 / 0.42 |
| Miyako 宮古特別地域気象観測所 | Miyako-city, Iwate-pref. 岩手県宮古市鍬ヶ崎下町 2-33 | 39°38.8'N 141°57.9'E | 42.5 | 1883 (1939[2], 1990[1]) | Robitzsch | No (2007-) | 1259.15 | 59,430 / 47 | 90,727 / N/A |
| Nemuro 根室特別地域気象観測所 | Nemuro-city, Hokkaido-pref. 北海道根室市弥栄町 1-18 | 43°19.8'N 145°35.1'E | 25.2 | 1879 (1886[1]) | Robitzsch | No (2010-) | 506.25 | 29,201 / 58 | 101,753 / 0.56 |
| Shimizu 清水特別地域気象観測所 | Tosashimizu-city, Kouchi-pref. 高知県土佐清水市足摺岬 605 | 32°43.3'N 133°00.6'E | 31 | 1940[1] (none) | Robitzsch | No (2007-) | 266.34 | 16,029 / 60 | 28,353 / 0.46 |
| Shionomisaki 潮岬特別地域気象観測所 | Kushimoto-town, Wakayama-pref. 和歌山県東牟婁郡串本町潮岬 3380-1 | 33°27.0'N 135°45.4'E | 67.5 | 1912[3] (2009[1]) | Robitzsch | No (2009-) | 135.67 | 18,249 / 135 | 30,329 / N/A |
| Wajima 輪島特別地域気象観測所 | Wajima-city, Ishikawa-pref. 石川県輪島市鳳至町畠田 99-3 | 37°23.4'N 136°53.7'E | 5.2 | 1896 (1921, 1929[1]) | Robitzsch | No (2010-) | 426.32 | 29,858 / 70 | 67,305 / 0.50 |

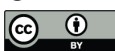

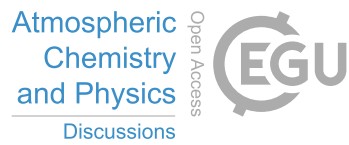

**Figure 2.** Information that characterizes the 14 observatories under study. Rows for the eight polluted and six pristine observatories are indicated in red and green, respectively. Columns for those directly related to the observatories and those describing the surroundings are indicated in yellow and blue, respectively. Listed for each observatory are the municipality, prefecture, and current address in Japanese (http://www.data.jma.go.jp/svd/eqev/data/kyoshin/jma_sindo.html) (note that station addresses are only partially made available on other websites), followed by the latitude, longitude, and altitude (m) of the current station location obtained

5  from the JMA website (http://www.data.jma.go.jp/obd/stats/etrn/index.php) and the years of foundation and relocation(s) extracted from the activity records of meteorological stations in Japan. It then indicates the type of radiation measurement instrument used during the 1960s (Eppley- or Robitzsch-type pyranometers), which had been in place before being replaced with thermopile pyranometer in early 1970s at all sites. The next column indicates whether staffs are resident at sites based on information available at the JMA website (http://www.data.jma.go.jp/obd/stats/data/kaisetu/shishin/shishin_3.pdf). Subsequently, this figure shows the municipality area (km$^2$) (http://www.gsi.go.jp/KOKUJYOHO/MENCHO/201410/opening.htm), 2010 census population count, population density (/km$^2$), vehicle $CO_2$

10  emissions (t$CO_2$/year) in 2003 (http://www.colgei.org/CO2/), and the number of registered vehicles (including light automobiles, so called K-cars) per capita in 2011 (Sangyochiiki_Kenkyusho, 2012). Vehicle $CO_2$ emissions are based on the number of registered vehicles in each municipality combined with assumptions on the average trip distance and the emission coefficient for each of several different vehicle types. Information shown above is for the current sites unless noted otherwise.

















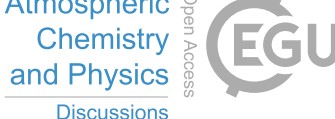



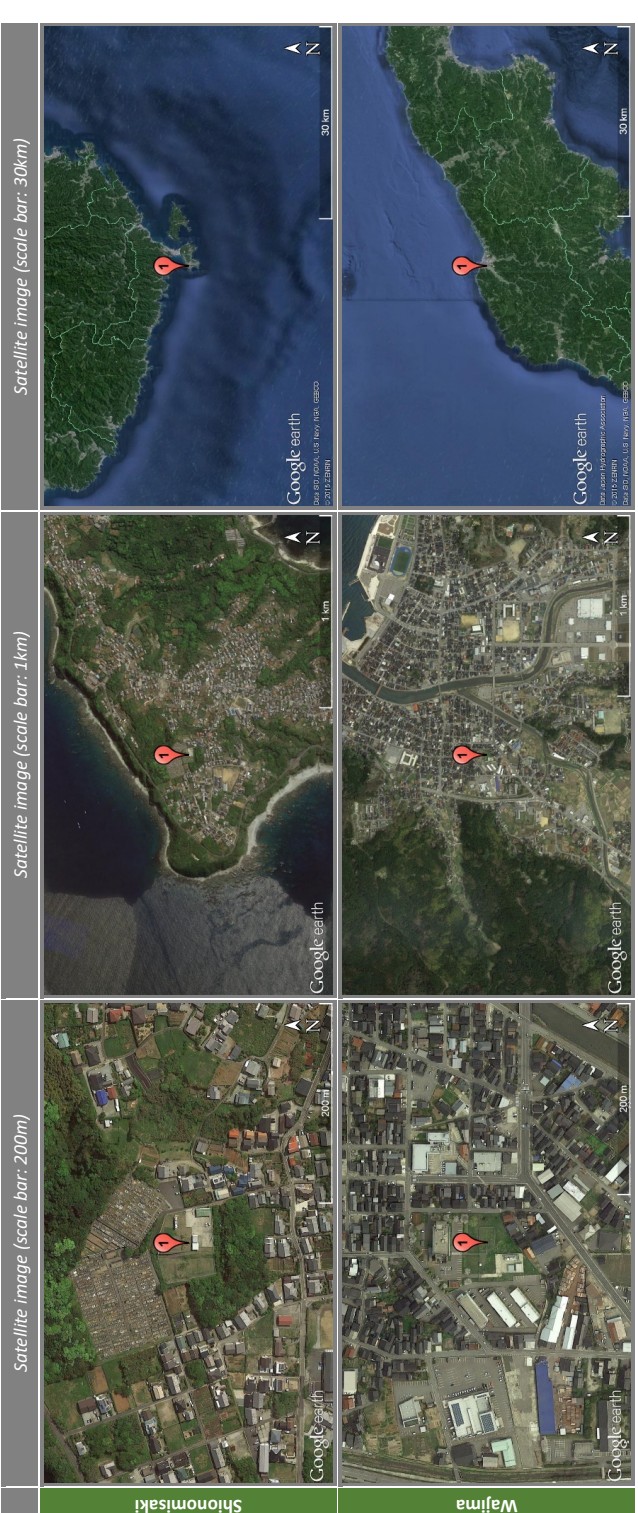

**Figure 3.** Satellite images surrounding the 14 stations under study. The heading for the eight polluted stations and the six pristine stations are indicated in red and green, respectively. Satellite images have been retrieved from Google Earth on 8 December 2015. The images are taken from satellites during 2012-2015. For each site, images are presented in three different scales: one showing the immediate surrounding such as buildings, housings, roads, and parks (scale bar = 200m), another showing the land use at the district level (scale bar = 1km), and the other covering largely the municipality area of interest (scale bar = 30km). The exact locations of the stations, which are marked as red balloons, are based on the postal addresses available at http://www.data.jma.go.jp/svd/eqev/data/kyoshin/jma_sindo.html. Green lines indicate current municipality boundaries; grey lines prefectural boundaries. It should be noted that the area surrounding the Miyako station has been heavily affected by the 2011 Tohoku Earthquake and Tsunami in Japan (see Figure 4 for images taken previously).



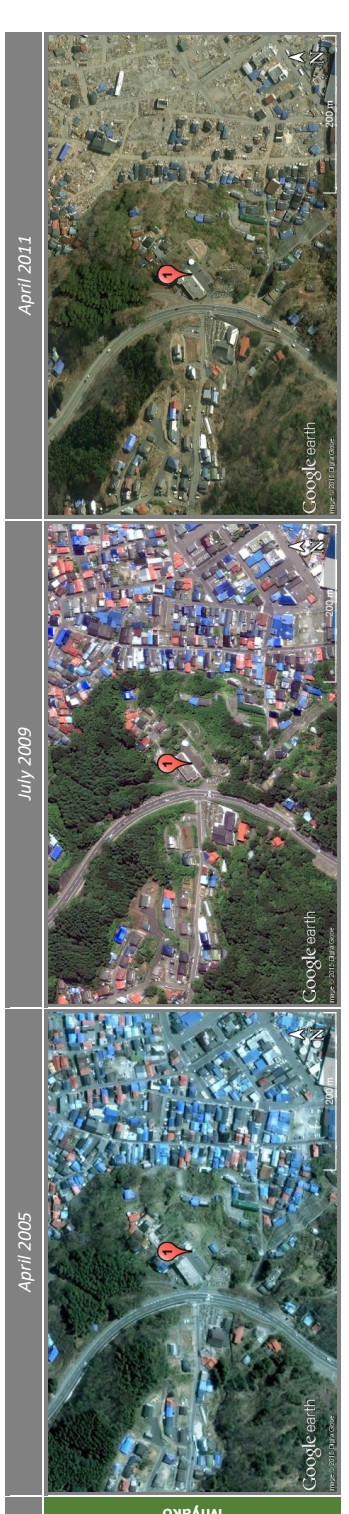

**Figure 4. Satellite images surrounding the Miyako station taken in April 2005, July 2009, and April 2011. Satellite images have been retrieved from Google Earth on 8 December 2015. The length of the scale bar is 200m. See the caption for Figure 3.**



















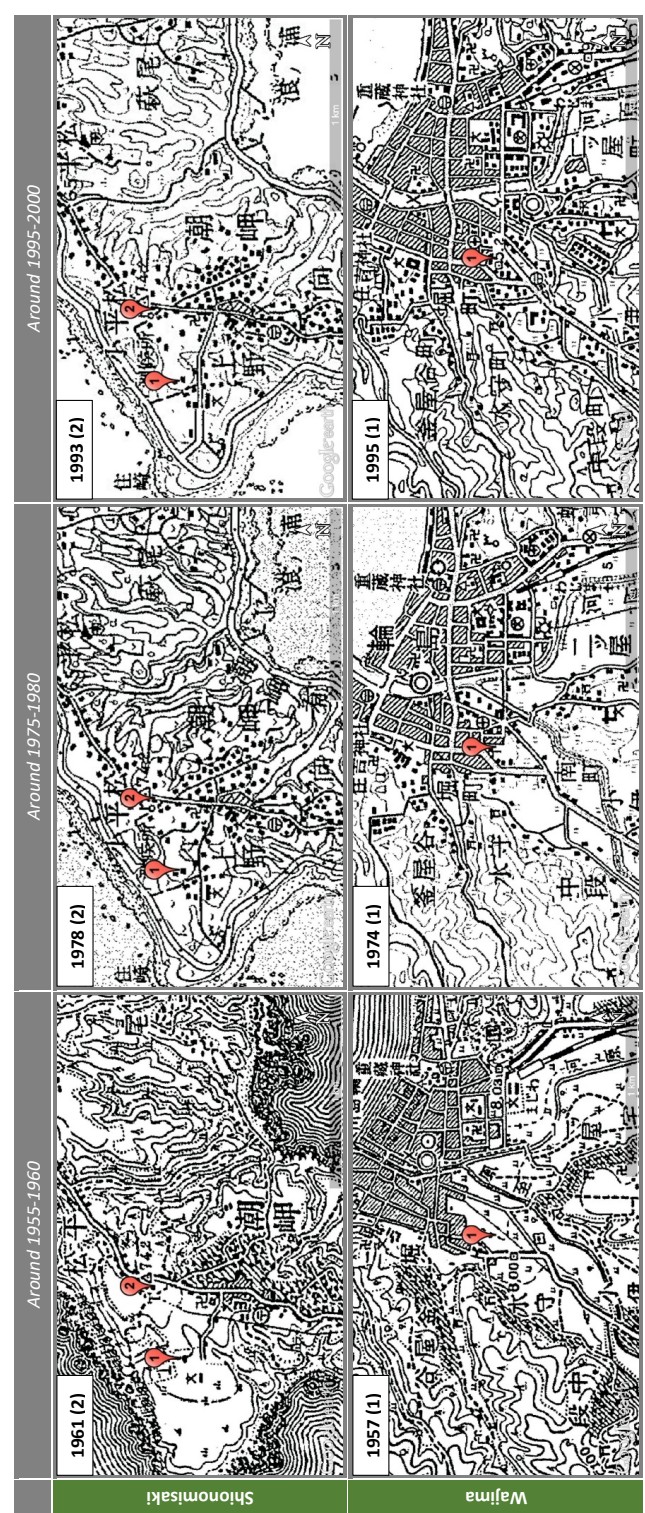

**Figure 5.** Chronological changes in land use surrounding the current and previous sites for the 14 observatories under study (length of the scale bar: 1km). At the top left corner of each panel, the issue year of the map is indicated. The number in bracket after the map year is the designated number of the site in operation at the time of map publication. Sites are numbered from the current one to the oldest up to around 1930s (see Figure 2 for the periods of operation). Site locations are approximately shown and mainly based on the activity records of meteorological observatories in Japan. Only substantial physical relocations accompanied by address changes are shown in the map. In other words, minor changes in instrument locations within the same premises are not made explicit in the figure. References for the maps used in this figure can be found in Figure 8.





















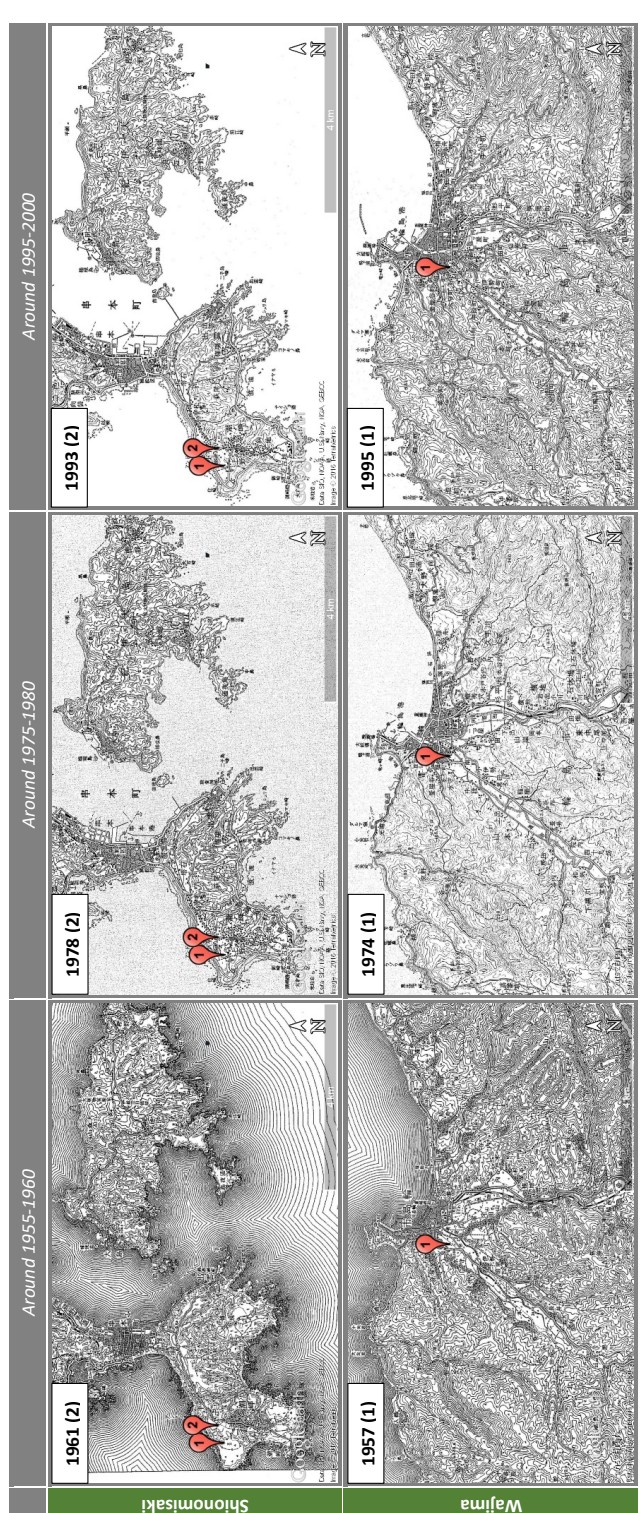

**Figure 6. Chronological changes in land use surrounding the current and previous sites for the 14 observatories under study (length of the scale bar: 4km). See the caption for Figure 5.**





**Figure 7. Land use changes at a few selected polluted sites (Akita, Fukuoka, and Matsumoto) in the early 20th century. The length of the scale bar is 4km. These maps can be compared with the corresponding ones in Figure 6. See the caption for Figure 5.**



| | |
|---|---|
| **Ishigakijima** | • Judicial branch of Government of the Ryukyu Islands, Ishigakijima [map]. 1:50,000. 1963.<br>• Geospatial Information Authority of Japan, Ishigakijima [map]. 1:50,000. 1974.<br>• Geospatial Information Authority of Japan, Ishigakijima [map]. 1:50,000. 1999. |
| **Miyako** | • Geographical Survey Institute, Miyako [map]. 1:50,000. 1957.<br>• Geospatial Information Authority of Japan, Miyako [map]. 1:50,000. 1976.<br>• Geospatial Information Authority of Japan, Miyako [map]. 1:50,000. 1993. |
| **Nemuro** | • Geographical Survey Institute, Nemuro north [map]. 1:50,000. 1957.<br>• Geospatial Information Authority of Japan, Nemuro south [map]. 1:50,000. 1960.<br>• Geospatial Information Authority of Japan, Nemuro north [map]. 1:50,000. 1976.<br>• Geospatial Information Authority of Japan, Nemuro north [map]. 1:50,000. 1977.<br>• Geospatial Information Authority of Japan, Nemuro north [map]. 1:50,000. 2002.<br>• Geospatial Information Authority of Japan, Nemuro south [map]. 1:50,000. 2003. |
| **Shimizu** | • Geographical Survey Institute, Tosashimizu [map]. 1:50,000. 1954.<br>• Geospatial Information Authority of Japan, Tosashimizu [map]. 1:50,000. 1973.<br>• Geospatial Information Authority of Japan, Tosashimizu [map]. 1:50,000. 1994. |
| **Shionomisaki** | • Geospatial Information Authority of Japan, Kushimoto [map]. 1:50,000. 1961.<br>• Geospatial Information Authority of Japan, Kushimoto [map]. 1:50,000. 1978.<br>• Geospatial Information Authority of Japan, Kushimoto [map]. 1:50,000. 1993. |
| **Wajima** | • Geographical Survey Institute, Wajima [map]. 1:50,000. 1957.<br>• Geospatial Information Authority of Japan, Wajima [map]. 1:50,000. 1974.<br>• Geospatial Information Authority of Japan, Wajima [map]. 1:50,000. 1995. |

**Figure 8.** References for the maps used in Figures 5, 6, and 7. Rows for the eight polluted and six pristine observatories are indicated in red and green, respectively.

| | |
|---|---|
| **Akita** | • Ordnance Survey Bureau of Imperial Japanese Army General Staff Office, Akita [map]. 1:50,000. 1926.<br>• Geographical Survey Institute, Akita [map]. 1:50,000. 1957.<br>• Geospatial Information Authority of Japan, Akita [map]. 1:50,000. 1977.<br>• Geospatial Information Authority of Japan, Akita [map]. 1:50,000. 1999. |
| **Fukuoka** | • Ordnance Survey Bureau of Imperial Japanese Army General Staff Office, Fukuoka [map]. 1:50,000. 1930.<br>• Geospatial Information Authority of Japan, Fukuoka [map]. 1:50,000. 1957.<br>• Geospatial Information Authority of Japan, Fukuoka [map]. 1:50,000. 1977.<br>• Geospatial Information Authority of Japan, Fukuoka [map]. 1:50,000. 1996. |
| **Kagoshima** | • Geographical Survey Institute, Kagoshima [map]. 1:50,000. 1954.<br>• Geospatial Information Authority of Japan, Kagoshima [map]. 1:50,000. 1977.<br>• Geospatial Information Authority of Japan, Kagoshima [map]. 1:50,000. 1993. |
| **Matsumoto** | • Ordnance Survey Bureau of Imperial Japanese Army General Staff Office, Matsumoto [map]. 1:50,000. 1918.<br>• Geographical Survey Institute, Matsumoto [map]. 1:50,000. 1959.<br>• Geospatial Information Authority of Japan, Matsumoto [map]. 1:50,000. 1976.<br>• Geospatial Information Authority of Japan, Matsumoto [map]. 1:50,000. 1995. |
| **Naha** | • Geographical Survey Institute, Wada [map]. 1:50,000. 1956.<br>• Geospatial Information Authority of Japan, Wada [map]. 1:50,000. 1976.<br>• Geospatial Information Authority of Japan, Wada [map]. 1:50,000. 1993.<br>• Judicial branch of Government of the Ryukyu Islands, Naha [map]. 1:50,000. 1962.<br>• Geospatial Information Authority of Japan, Naha [map]. 1:50,000. 1977.<br>• Geospatial Information Authority of Japan, Naha [map]. 1:50,000. 1996. |
| **Sapporo** | • Geographical Survey Institute, Sapporo [map]. 1:50,000. 1958.<br>• Geospatial Information Authority of Japan, Sapporo [map]. 1:50,000. 1977.<br>• Geospatial Information Authority of Japan, Sapporo [map]. 1:50,000. 2000. |
| **Taten** | • Geospatial Information Authority of Japan, Tsuchiura [map]. 1:50,000. 1960.<br>• Geospatial Information Authority of Japan, Tsuchiura [map]. 1:50,000. 1977.<br>• Geospatial Information Authority of Japan, Tsuchiura [map]. 1:50,000. 1998. |
| **Yonago** | • Geographical Survey Institute, Yonago [map]. 1:50,000. 1957.<br>• Geospatial Information Authority of Japan, Yonago [map]. 1:50,000. 1977.<br>• Geospatial Information Authority of Japan, Yonago [map]. 1:50,000. 1995. |













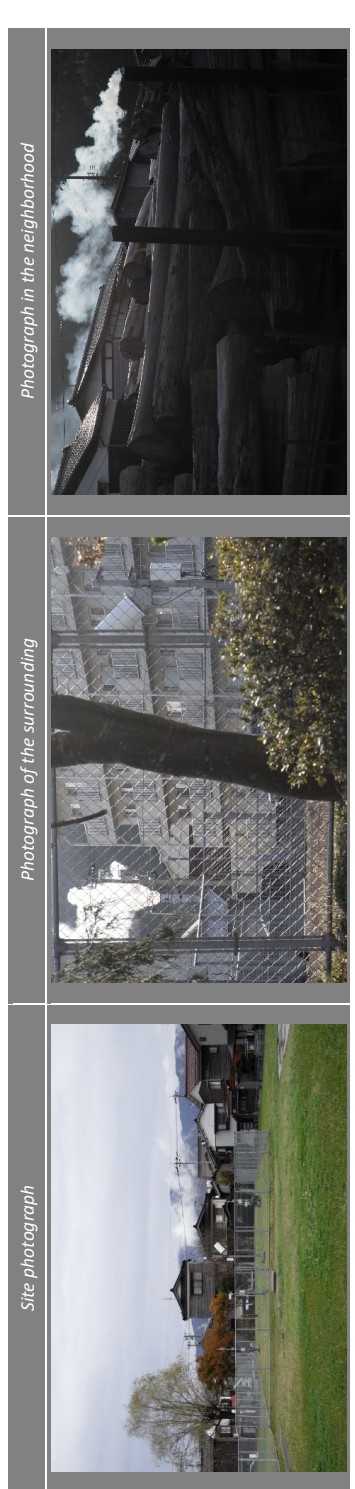

**Figure 9. Selected photographs taken during the site visits by K. Tanaka. He paid visit to Akita on 22 November 2015; Matsumoto on 19 November 2015; Tateno on 25 December 2015; Wajima on 20 November 2015 and 28 March 2016; Yonago on 3 January 2016; Shimizu on 29 December 2015; and; Shionomisaki on 3 November 2015 and 3 May 2016.**





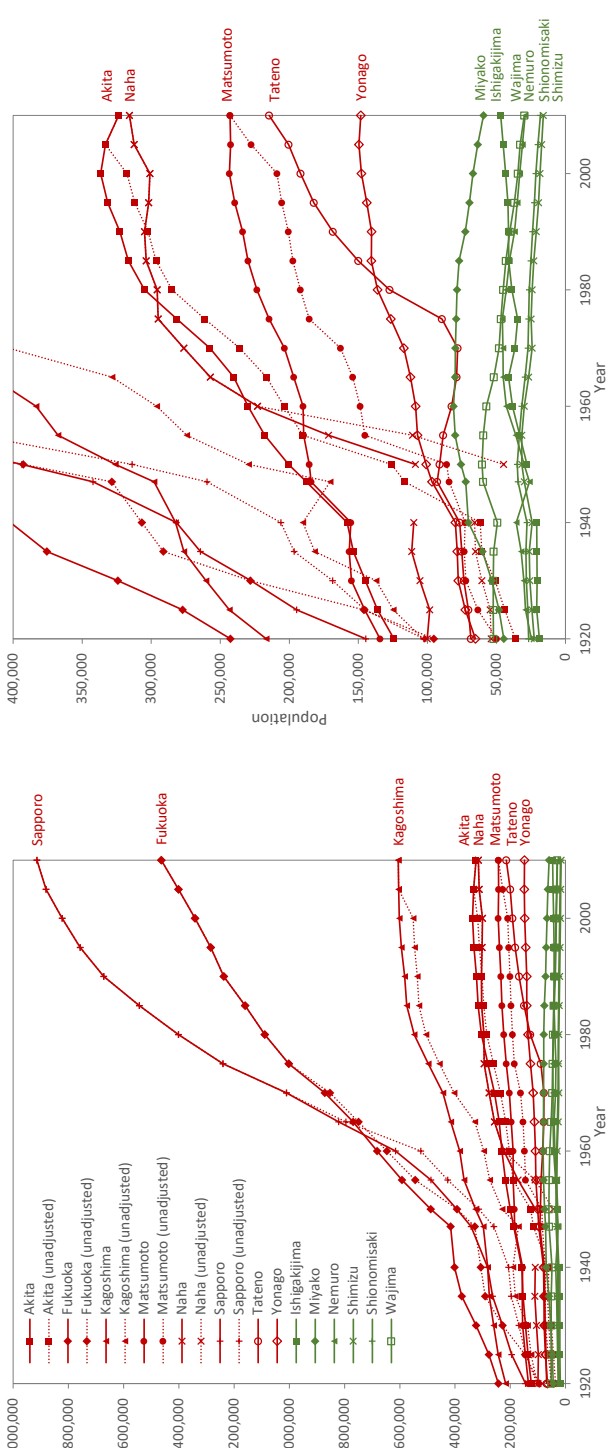

**Figure 10.** Changes in the census population since 1920 at the municipalities where the **14 Japanese stations are located (in two different vertical scales)**. Red and green lines indicate polluted and pristine stations, respectively. There have been a number of changes in the municipality boundaries in Japan in particular during the past 15 years. Population data in solid lines are based on consistent boundaries – in other words, current boundaries are kept all the way back to 1920 to derive population changes. In contrast, population data in dotted line show those without such corrections after 1980. The nation-wide census takes place every five years since 1920, with the 1947 census as a replacement with one in 1945 being an exception. There are data gaps in the population data for the Naha and Ishigakijima stations in Okinawa prefecture in 1947 as a result of the World War II.



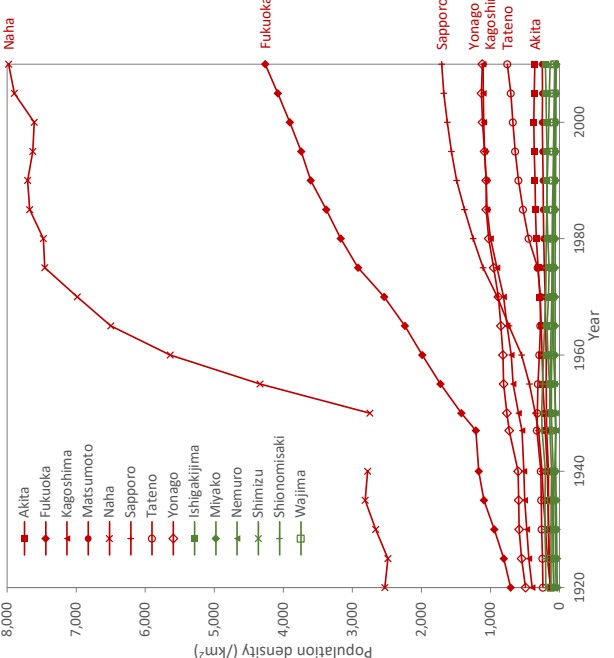

**Figure 11. Changes in the population density since 1920 at the municipalities where the 14 Japanese stations are located (in two different vertical scales). See the caption for Figure 10.**





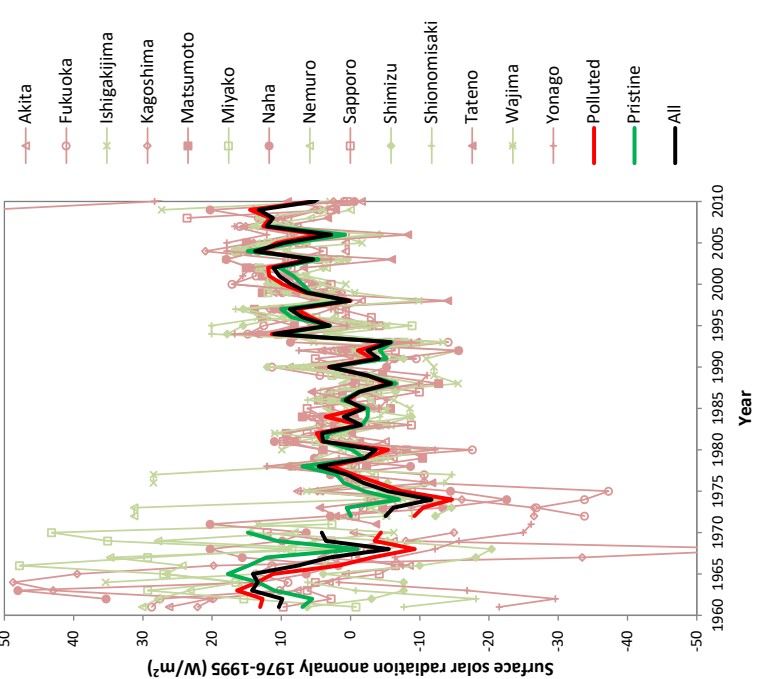

**Figure 12. Surface solar radiation anomalies of the 14 Japanese stations used in our analysis. The thick red, green, and black lines are the averages for polluted, pristine, and all stations, respectively. The figure shows the means for SSR and zenith/maximum transmittance data over the polluted, pristine, and all stations only if there are equal to or more than three, three, and six data points, respectively.**



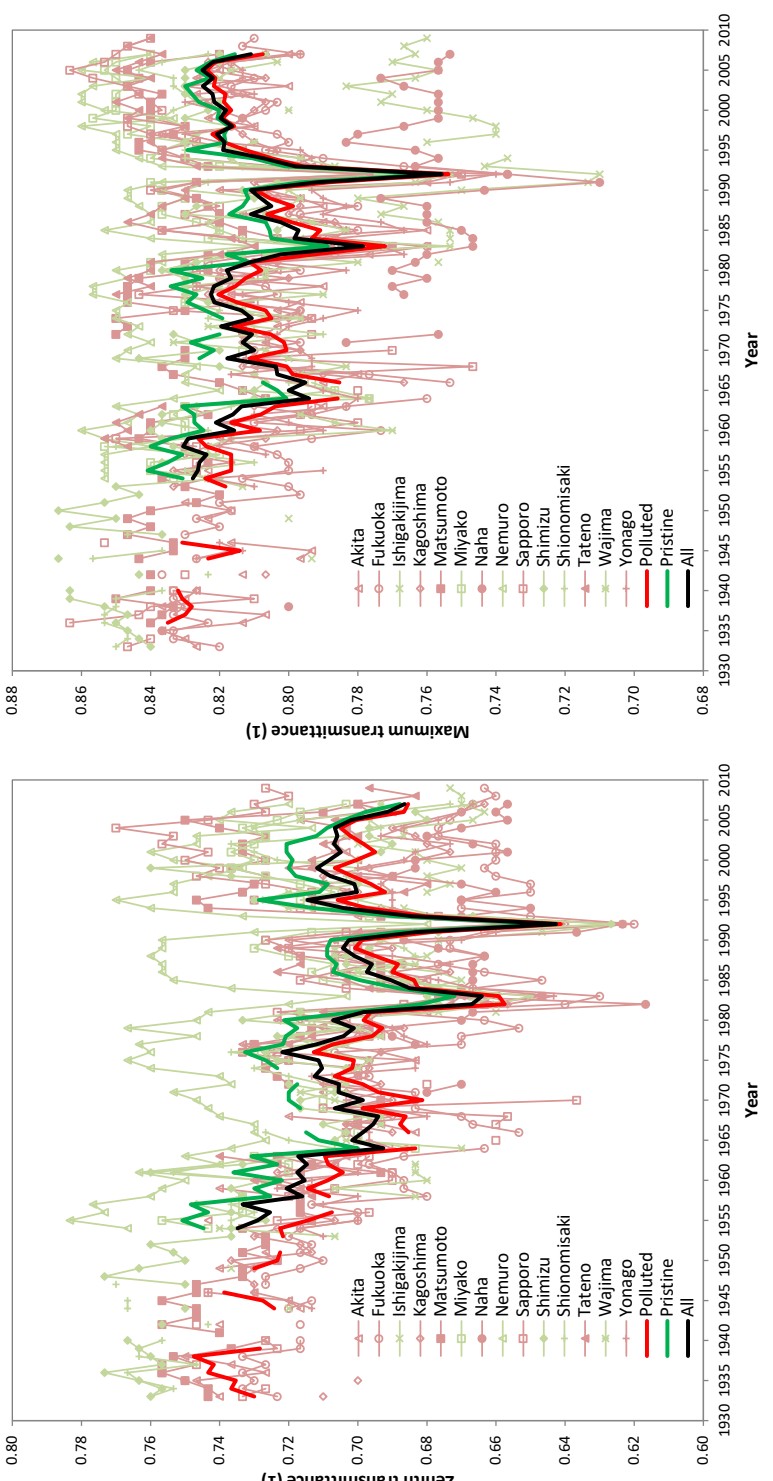

**Figure 13. Zenith transmittance and maximum transmittance of the 14 Japanese stations used in our analysis. See the caption for Figure 12.**







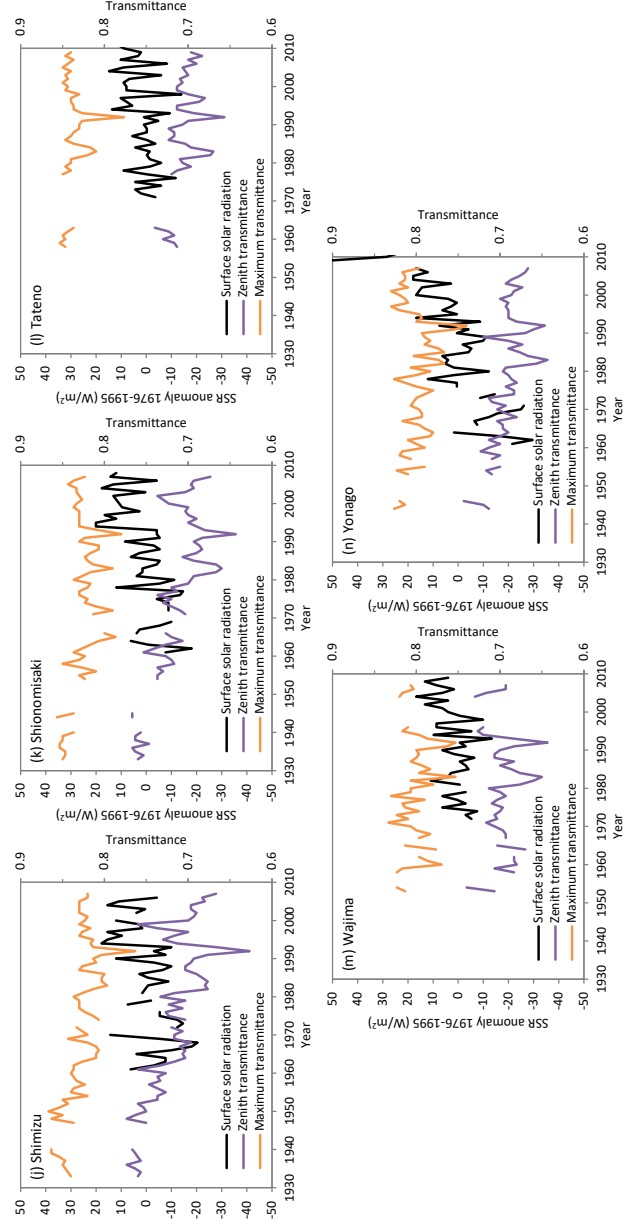

**Figure 14. Surface solar radiation anomalies, zenith transmittance, and maximum transmittance measured at individual stations.**







**Figure 15. Surface solar radiation anomalies of the 14 Japanese stations categorized into groups in alternative ways. In the sensitivity cases above, the Ishigakijima and Wajima stations, which is in the pristine group in the reference case, are switched to the polluted group one by one (cases I and III) and altogether (case II). See the caption and legend for Figure 12.**




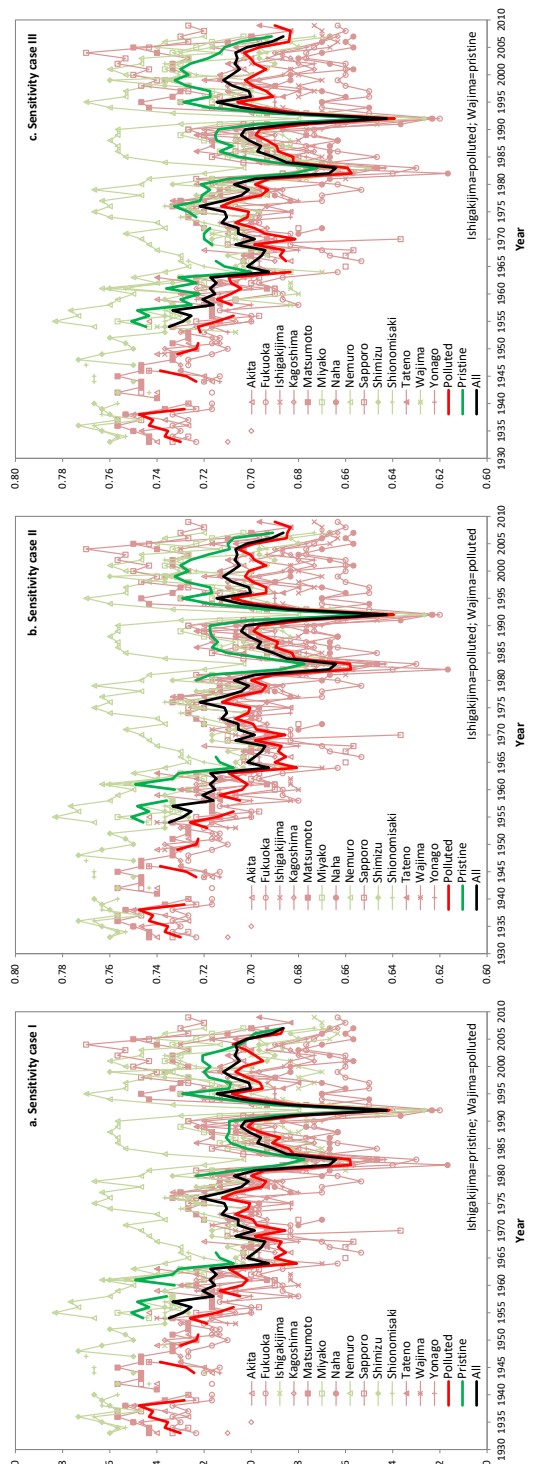

**Figure 16. Zenith transmittance of the 14 Japanese stations categorized into groups in alternative ways. See the caption for Figure 15.**





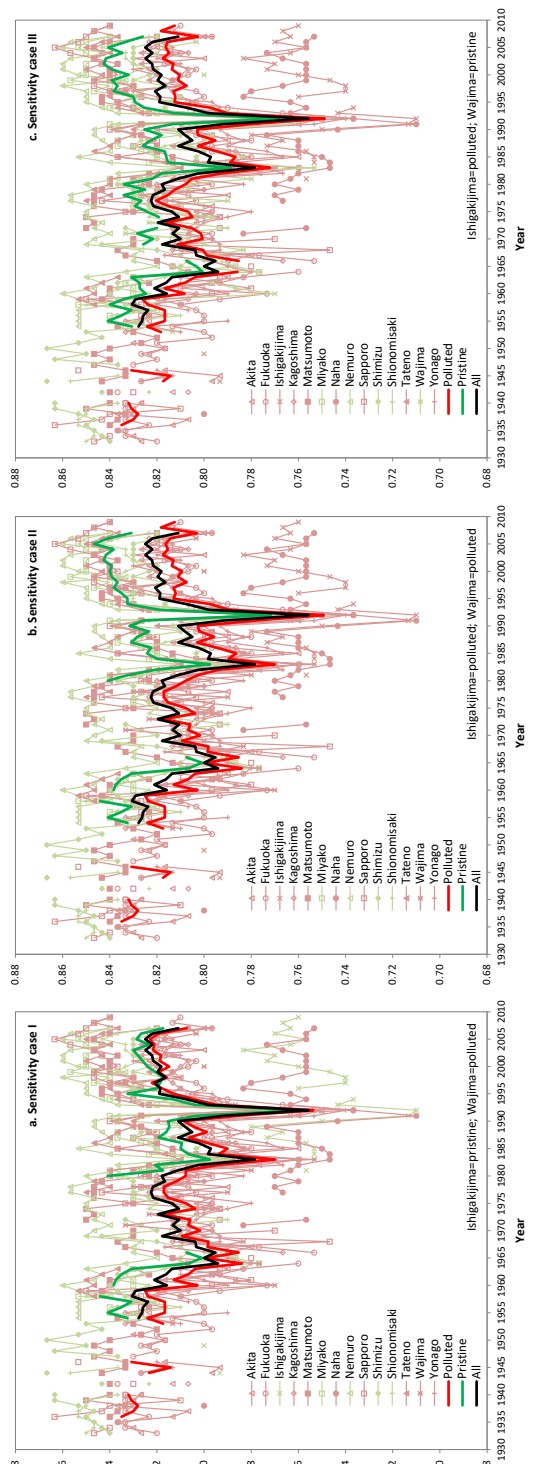

**Figure 17. Maximum transmittance of the 14 Japanese stations categorized into groups in alternative ways. See the caption for Figure 15.**