# Peer review of "Is global dimming and brightening in Japan limited to urban areas?"

_Atmospheric Chemistry and Physics, 2016_

## Referee Comment (RC1) · Anonymous Referee #1 · 29 Aug 2016

The paper describes that the decadal trend of the surface solar radiation (global dimming and brightening) in Japan is a phenomenon by local air pollution or the large scale changes in background aerosols. This is an important point to investigate a cause of the global dimming and brightening. The authors separate the observatories into the polluted and pristine stations by carefully checking the historical land use map, population time series, satellite image, and actual site visits. This approach is reliable and would be useful in studying the cases in the other countries. The paper is generally well written and fits the scope of ACP. I think the paper could be published after minor changes.

Technical comments:

1. Check all the in-text citations. For example, "(Ohmura and Lang, 1989)" in P2, L3

should be written by "Ohmura and Lang (1989)".

2. Figs. 3, 4, 5, 6, and 7. The scale bars are not clear.

3. Figs. 5, 6, and 7. The legends indicating the urbanization (e.g., residential area) are necessary. 4. P15, L21-28. The references of Ramanathan 2007a and 2007b are same.

General comments:

1. P3, L24-30. The approach using both the population data and the land use map is a reliable method. However, if the population data is enough to show the urbanization, it is useful to classify the observatories worldwide into the polluted and pristine stations. The change in the population is strongly related to the land use. The population does not increase without the increases of the residential area and the commercial facilities. Is the land use map necessary in the classification?

2. P11, L19-24. The trends of the transmittance are in line with those in the surface solar radiation. I would like to know the change in the transmittance can explain the changes in the surface solar radiation quantitatively. Could you estimate the change in the surface solar radiation from the change in the transmittance? I think that the direct component of the surface solar radiation can be estimated roughly but it is difficult to estimate the diffuse component.

The annual means of the transmittance are calculated from the data under the clear sky condition. Such data may not be a representative of aerosols in a year. Please show how many days of the transmittance data are used to calculate the annual means.

---

## Referee Comment (RC2) · Anonymous Referee #2 · 8 Oct 2016

Whether the observed global diming and brightening of surface solar radiation is limited to urban areas? This is an important but not well solved issue. Different from other studies, the authors made great effort in collecting meta-data of the observations, including different indices related to urbanization and information on the instruments. The information is important for related studies. This paper can be a model to be followed by the similar studies in other countries. I strongly recommend the author's efforts and publication of this paper. I found minors errors but have already mentioned by the first referee.

---

## Author Response (AR1)

**NATIONAL INSTITUTE FOR ENVIRONMENTAL STUDIES**
CENTER FOR GLOBAL ENVIRONMENTAL RESEARCH

**DR. KATSUMASA TANAKA**
16-2 Onogawa, 305-8506 Tsukuba, JAPAN
Phone:+81-29-850-2493
tanaka.katsumasa@nies.go.jp
http://www.nies.go.jp

*Copernicus Gesellschaft mbH*
*Bahnhofsallee 1e*
*37081 Göttingen*
*Germany*
17 October 2016

Dear Prof. Toshihiko Takemura,

Please find enclosed our revised manuscript, "Is global dimming and brightening in Japan limited to urban areas?" by K. Tanaka, A. Ohmura, D. Folini, M. Wild, and N. Ohkawara, which we would like to submit to be considered for publication as a Research Article in *Atmospheric Chemistry and Physics (ACP)*.

We have received a set of two referee reports, both of which are generally positive. Referee #1 suggested a publication after minor changes. Referee #2 endorsed our paper by stating "This paper can be a model to be followed by the similar studies in other countries. I strongly recommend the author's efforts and publication of this paper." Given such feedback, we keep the manuscript as it was, except for minor changes suggested by Referee #1. Changes that have been made on the manuscript are indicated in our point-by-point responses to referees' comments.

Our manuscript has 6,381 words (main text), 207 words (abstract), 17 figures, and 70 references. Thank you for considering our manuscript for publication in *ACP*. We look forward to receiving your feedback.

Yours sincerely,

Katsumasa Tanaka

**NATIONAL INSTITUTE FOR ENVIRONMENTAL STUDIES**
CENTER FOR GLOBAL ENVIRONMENTAL RESEARCH

**DR. KATSUMASA TANAKA**
16-2 Onogawa, 305-8506 Tsukuba, JAPAN
Phone:+81-29-850-2493
tanaka.katsumasa@nies.go.jp
http://www.nies.go.jp

**To Referee #1**

[Referee's comment] *The paper describes that the decadal trend of the surface solar radiation (global dimming and brightening) in Japan is a phenomenon by local air pollution or the large scale changes in background aerosols. This is an important point to investigate a cause of the global dimming and brightening. The authors separate the observatories into the polluted and pristine stations by carefully checking the historical land use map, population time series, satellite image, and actual site visits. This approach is reliable and would be useful in studying the cases in the other countries. The paper is generally well written and fits the scope of ACP. I think the paper could be published after minor changes.*

[Our response] Thank you for reviewing our paper and providing us with detailed comments. All the comments were helpful to refine our manuscript. Please see our point-by-point responses to your comments below.

*Technical comments:*
[Referee's comment] *1. Check all the in-text citations. For example, "(Ohmura and Lang, 1989)" in P2, L3 should be written by "Ohmura and Lang (1989)".*

[Our response] We have converted all the in-text citations to the proper form.

[Referee's comment] *2. Figs. 3, 4, 5, 6, and 7. The scale bars are not clear.*

[Our response] We generated these figures by using Google Earth Pro. The size of scale bars was already determined to what we thought optimal. As far as we are aware, there is little room to do further within Google Earth Pro. To improve the scale bar visibility in the revised manuscript, we thus worked directly on the images by adjusting their contrast and sharpness. The end results are somewhat better. Furthermore, if this paper is published in *Atmospheric Chemistry and Physics*, the journal might be able to offer larger printing space than that in *Atmospheric Chemistry and Physics Discussions*, in which these figures could be shown larger, which would help make the scale bars more visible.

[Referee's comment] *3. Figs. 5, 6, and 7. The legends indicating the urbanization (e.g., residential area) are necessary.*

[Our response] We are afraid that we do not understand what exactly this particular comment suggested. It may be unclear what the red and green rows mean when these figures are inspected alone. In the caption of Figure 5, we thus added the following sentence: "Rows for the eight polluted and six pristine observatories are indicated in red and green, respectively." We did not add this sentence to Figures 6 and 7 because the captions for Figures 6 and 7 refer to that for Figure 5.

[Figure]

NATIONAL INSTITUTE FOR ENVIRONMENTAL STUDIES
CENTER FOR GLOBAL ENVIRONMENTAL RESEARCH

**DR. KATSUMASA TANAKA**
16-2 Onogawa, 305-8506 Tsukuba, JAPAN
Phone:+81-29-850-2493
tanaka.katsumasa@nies.go.jp
http://www.nies.go.jp

[Referee's comment] *4. P15, L21-28. The references of Ramanathan 2007a and 2007b are same.*

[Our response] Thank you for pointing this out. We did not notice this duplicate. These two references are merged in the revised manuscript.

*General comments:*
[Referee's comment] *1. P3, L24-30. The approach using both the population data and the land use map is a reliable method. However, if the population data is enough to show the urbanization, it is useful to classify the observatories worldwide into the polluted and pristine stations. The change in the population is strongly related to the land use. The population does not increase without the increases of the residential area and the commercial facilities. Is the land use map necessary in the classification?*

[Our response] This comment is related to one of the very motivations for this study. It is indeed a common practice to use population data as a proxy for urbanization. However, in the debate on the possible causes of global dimming and brightening phenomenon, this approach is questioned. Our manuscript contains the following paragraph in the method section, which we hope clarities your concern:

"Population data are the most accessible and only available long time-series on such a fine scale (note that municipality population data before 1970s are not systematically available in a digitalized format). There are, however, known limitations when they are interpreted as proxies for urbanization. The use of population alone as a proxy to infer the air pollution level is under debate (Alpert et al., 2005; Ramanathan et al., 2007; Alpert and Kishcha, 2008; Stanhill and Cohen, 2009; Wang et al., 2014; Imamovic et al., 2016). Related are issues associated with the use of population-based proxies to infer $CO_2$ emissions (for an overview, see Andres et al. (2012)), even though $CO_2$ emissions are not necessarily correlated with $SO_2$ emissions on the relevant spatial and time scales (Grossman and Krueger, 1995; Holtz-Eakin and Selden, 1995; Dinda, 2004). Population data may serve as a proxy for $CO_2$ emissions from residential and commercial sectors but work poorly for emissions from power and transport sectors. With respect to power sectors, population fails to represent $CO_2$ emissions when coal is combusted in a remote area to support the electricity demands in a distant urban area. This type of issues becomes important when one deals with emission data on a fine spatial scale. Concerning transport sectors, transport emissions per capita are known to decline above a certain population density threshold (Gately et al., 2015). In our analysis, the population time-series are therefore complemented by the series of historical land use maps, recent satellite images, and current photographs, which provide additional insights into how the site surroundings have been changed during the past period of interest."

[Figure]

**NATIONAL INSTITUTE FOR ENVIRONMENTAL STUDIES**
CENTER FOR GLOBAL ENVIRONMENTAL RESEARCH

**DR. KATSUMASA TANAKA**
16-2 Onogawa, 305-8506 Tsukuba, JAPAN
Phone:+81-29-850-2493
tanaka.katsumasa@nies.go.jp
http://www.nies.go.jp

To emphasize this argument upfront in this paper, we revised the following sentence in line 24 on page 3 (the underlined part was added in the revised manuscript):

 "While a few earlier studies (e.g. Alpert et al., 2005; Alpert and Kishcha, 2008) rely primarily on population data to infer the influence of urbanization on SSR measurements, we collect and utilize as much information as possible to overcome possible limitations for population-based proxies when they are applied to infer the air pollution level (Ramanathan et al., 2007; Stanhill and Cohen, 2009; Wang et al., 2014; Imamovic et al., 2016)."

[Referee's comment] *2. P11, L19-24. The trends of the transmittance are in line with those in the surface solar radiation. I would like to know the change in the transmittance can explain the changes in the surface solar radiation quantitatively. Could you estimate the change in the surface solar radiation from the change in the transmittance? I think that the direct component of the surface solar radiation can be estimated roughly but it is difficult to estimate the diffuse component.*

[Our response] Such computations would require a radiative transfer model like MODTRAN. However, using such a model would lead to another new study. For this paper, we would thus stop here without going into such numerical computations.

We appreciate this suggestion from a general point of view. We in fact did not analyze the data in a quantitatively rigorous way. We rather focused on the collection of data (in terms of both the quantity and quality), which itself is a project, and we interpreted them mostly by visual inspection. Some of the collected data (e.g. maps) cannot be even quantified. Nevertheless, the data we collected offer a new opportunity for more quantitative analyses, which is a future research direction.

[Referee's comment] *The annual means of the transmittance are calculated from the data under the clear sky condition. Such data may not be a representative of aerosols in a year. Please show how many days of the transmittance data are used to calculate the annual means.*

[Our response] When there are more than six months of transmission data available within a year, we take the mean and plot it as an annual-mean value. The threshold of six months is arbitrary. Another way would have been to fill the missing 6 months with the climatological mean months that can be determined from the same site. This would exclude the possibility that biases can be introduced in the annual means by not sampling a part of the year (which for example could be on average higher or lower than the 6 months available). However, we don't think it feasible to implement this at this stage of the review process.

[Figure]

**NATIONAL INSTITUTE FOR ENVIRONMENTAL STUDIES**
CENTER FOR GLOBAL ENVIRONMENTAL RESEARCH

**DR. KATSUMASA TANAKA**
16-2 Onogawa, 305-8506 Tsukuba, JAPAN
Phone:+81-29-850-2493
tanaka.katsumasa@nies.go.jp
http://www.nies.go.jp

The original manuscript was not explicit about how we arrived at the annual-mean transmission data. We thus add the following sentence to the revised manuscript in line 15 on page 10:

"When there are more than six months of transmission data available within a year, we take the mean and use it as an annual-mean value."

[Figure]

**NATIONAL INSTITUTE FOR ENVIRONMENTAL STUDIES**
CENTER FOR GLOBAL ENVIRONMENTAL RESEARCH

**DR. KATSUMASA TANAKA**
16-2 Onogawa, 305-8506 Tsukuba, JAPAN
Phone:+81-29-850-2493
tanaka.katsumasa@nies.go.jp
http://www.nies.go.jp

**To Referee #2**

[Referee's comment] *Whether the observed global diming and brightening of surface solar radiation is limited to urban areas? This is an important but not well solved issue. Different from other studies, the authors made great effort in collecting meta-data of the observations, including different indices related to urbanization and information on the instruments. The information is important for related studies. This paper can be a model to be followed by the similar studies in other countries. I strongly recommend the author's efforts and publication of this paper. I found minors errors but have already mentioned by the first referee.*

[Our response] Thank you for reviewing our paper and providing us with the constructive comment. We appreciate your strong support for our study. All minor errors by Referee #1 are addressed in the revised manuscript.

[revised manuscript text omitted]

**Figure 5. Chronological changes in land use surrounding the current and previous sites for the 14 observatories under study (length of the scale bar: 1km). Rows for the eight polluted and six pristine observatories are indicated in red and green, respectively. At the top left corner of each panel, the issue year of the map is indicated. The number in bracket after the map year is the designated number of the site in operation at the time of map publication. Sites are numbered from the current one to the oldest up to around 1930s (see Figure 2 for the periods of operation). Site locations are approximately shown and mainly based on the activity records of meteorological observatories in Japan. Only substantial physical relocations accompanied by address changes are shown in the map. In other words, minor changes in instrument locations within the same premises are not made explicit in the figure. References for the maps used in this figure can be found in Figure 8.**

[Figure]

| | *Around 1955-1960* | *Around 1975-1980* | *Around 1995-2000* |
|---|---|---|---|
| **Akita** | 1957 (2) | 1977 (2) | 1999 (1) |
| **Fukuoka** | 1957 (1) | 1977 (1) | 1996 (1) |
| **Kagoshima** | 1954 (2) | 1977 (2) | 1993 (2) |

[Figure]

[Figure]

[Figure]

|  | Around 1955-1960 | Around 1975-1980 | Around 1995-2000 |
|---|---|---|---|
| Miyako | 1957 (2) | 1976 (2) | 1993 (1) |
| Nemuro | 1960 (1) | 1977 (1) | 2003 (1) |
| Shimizu | 1954 (1) | 1968 (1) | 1994 (1) |

[Figure]

**Figure 6. Chronological changes in land use surrounding the current and previous sites for the 14 observatories under study (length of the scale bar: 4km). See the caption for Figure 5.**

[Figure]

**Figure 7. Land use changes at a few selected polluted sites (Akita, Fukuoka, and Matsumoto) in the early 20th century. The length of the scale bar is 4km. These maps can be compared with the corresponding ones in Figure 6. See the caption for Figure 5.**

| | |
|---|---|
| **Akita** | • Ordnance Survey Bureau of Imperial Japanese Army General Staff Office, Akita [map]. 1:50,000. 1926.
• Geographical Survey Institute, Akita [map]. 1:50,000. 1957.
• Geospatial Information Authority of Japan, Akita [map]. 1:50,000. 1977.
• Geospatial Information Authority of Japan, Akita [map]. 1:50,000. 1999. |
| **Fukuoka** | • Ordnance Survey Bureau of Imperial Japanese Army General Staff Office, Fukuoka [map]. 1:50,000. 1930.
• Geospatial Information Authority of Japan, Fukuoka [map]. 1:50,000. 1957.
• Geospatial Information Authority of Japan, Fukuoka [map]. 1:50,000. 1977.
• Geospatial Information Authority of Japan, Fukuoka [map]. 1:50,000. 1996. |
| **Kagoshima** | • Geographical Survey Institute, Kagoshima [map]. 1:50,000. 1954.
• Geospatial Information Authority of Japan, Kagoshima [map]. 1:50,000. 1977.
• Geospatial Information Authority of Japan, Kagoshima [map]. 1:50,000. 1993. |
| **Matsumoto** | • Ordnance Survey Bureau of Imperial Japanese Army General Staff Office, Matsumoto [map]. 1:50,000. 1918.
• Geographical Survey Institute, Matsumoto [map]. 1:50,000. 1959.
• Geospatial Information Authority of Japan, Matsumoto [map]. 1:50,000. 1976.
• Geospatial Information Authority of Japan, Matsumoto [map]. 1:50,000. 1995.
• Geographical Survey Institute, Wada [map]. 1:50,000. 1956.
• Geospatial Information Authority of Japan, Wada [map]. 1:50,000. 1976.
• Geospatial Information Authority of Japan, Wada [map]. 1:50,000. 1993. |
| **Naha** | • Judicial branch of Government of the Ryukyu Islands, Naha [map]. 1:50,000. 1962.
• Geospatial Information Authority of Japan, Naha [map]. 1:50,000. 1977.
• Geospatial Information Authority of Japan, Naha [map]. 1:50,000. 1996. |
| **Sapporo** | • Geographical Survey Institute, Sapporo [map]. 1:50,000. 1958.
• Geospatial Information Authority of Japan, Sapporo [map]. 1:50,000. 1977.
• Geospatial Information Authority of Japan, Sapporo [map]. 1:50,000. 2000. |
| **Taten** | • Geospatial Information Authority of Japan, Tsuchiura [map]. 1:50,000. 1960.
• Geospatial Information Authority of Japan, Tsuchiura [map]. 1:50,000. 1977.
• Geospatial Information Authority of Japan, Tsuchiura [map]. 1:50,000. 1998. |
| **Yonago** | • Geographical Survey Institute, Yonago [map]. 1:50,000. 1957.
• Geospatial Information Authority of Japan, Yonago [map]. 1:50,000. 1977.
• Geospatial Information Authority of Japan, Yonago [map]. 1:50,000. 1995. |

| | |
|---|---|
| **Ishigakijima** | • Judicial branch of Government of the Ryukyu Islands, Ishigakijima [map]. 1:50,000. 1963.
• Geospatial Information Authority of Japan, Ishigakijima [map]. 1:50,000. 1974.
• Geospatial Information Authority of Japan, Ishigakijima [map]. 1:50,000. 1999. |
| **Miyako** | • Geographical Survey Institute, Miyako [map]. 1:50,000. 1957.
• Geospatial Information Authority of Japan, Miyako [map]. 1:50,000. 1976.
• Geospatial Information Authority of Japan, Miyako [map]. 1:50,000. 1993. |
| **Nemuro** | • Geographical Survey Institute, Nemuro north [map]. 1:50,000. 1957.
• Geospatial Information Authority of Japan, Nemuro south [map]. 1:50,000. 1960.
• Geospatial Information Authority of Japan, Nemuro north [map]. 1:50,000. 1976.
• Geospatial Information Authority of Japan, Nemuro south [map]. 1:50,000. 1977.
• Geospatial Information Authority of Japan, Nemuro north [map]. 1:50,000. 2002.
• Geospatial Information Authority of Japan, Nemuro south [map]. 1:50,000. 2003. |
| **Shimizu** | • Geographical Survey Institute, Tosashimizu [map]. 1:50,000. 1954.
• Geospatial Information Authority of Japan, Tosashimizu [map]. 1:50,000. 1973.
• Geospatial Information Authority of Japan, Tosashimizu [map]. 1:50,000. 1994. |
| **Shionomisaki** | • Geospatial Information Authority of Japan, Kushimoto [map]. 1:50,000. 1961.
• Geospatial Information Authority of Japan, Kushimoto [map]. 1:50,000. 1978.
• Geospatial Information Authority of Japan, Kushimoto [map]. 1:50,000. 1993. |
| **Wajima** | • Geographical Survey Institute, Wajima [map]. 1:50,000. 1957.
• Geospatial Information Authority of Japan, Wajima [map]. 1:50,000. 1974.
• Geospatial Information Authority of Japan, Wajima [map]. 1:50,000. 1995. |

**Figure 8. References for the maps used in Figures 5, 6, and 7. Rows for the eight polluted and six pristine observatories are indicated in red and green, respectively.**

[Figure]

[Figure]

|  | Site photograph | Photograph of the surrounding | Photograph in the neighborhood |
| --- | --- | --- | --- |
| Yonago | | | |
| Shimizu | | | |
| Shionomisaki | | | |

| | Site photograph | Photograph of the surrounding | Photograph in the neighborhood |
|---|---|---|---|
| Wajima |
[Figure]
 | | |

**Figure 9. Selected photographs taken during the site visits by K. Tanaka. He paid visit to Akita on 22 November 2015; Matsumoto on 19 November 2015; Tateno on 25 December 2015; Wajima on 20 November 2015 and 28 March 2016; Yonago on 3 January 2016; Shimizu on 29 December 2015, and; Shionomisaki on 3 November 2015 and 3 May 2016.**

[Figure]

**Figure 10. Changes in the census population since 1920 at the municipalities where the 14 Japanese stations are located (in two different vertical scales). Red and green lines indicate polluted and pristine stations, respectively. There have been a number of changes in the municipality boundaries in Japan in particular during the past 15 years. Population data in solid lines are based on consistent boundaries – in other words, current boundaries are kept all the way back to 1920 to derive population changes. In contrast, population data in dotted line show those without such corrections after 1980. The nation-wide census takes place every five years since 1920, with the 1947 census as a replacement with one in 1945 being an exception. There are data gaps in the population data for the Naha and Ishigakijima stations in Okinawa prefecture in 1947 as a result of the World War II.**

[Figure]

**Figure 11. Changes in the population density since 1920 at the municipalities where the 14 Japanese stations are located (in two different vertical scales). See the caption for Figure 10.**

[Figure]

**Figure 12. Surface solar radiation anomalies of the 14 Japanese stations used in our analysis. The thick red, green, and black lines are the averages for polluted, pristine, and all stations, respectively. The figure shows the means for SSR and zenith/maximum transmittance data over the polluted, pristine, and all stations only if there are equal to or more than three, three, and six data points, respectively.**

[Figure]

**Figure 13. Zenith transmittance and maximum transmittance of the 14 Japanese stations used in our analysis. See the caption for Figure 12.**

[Figure]

[Figure]

**Figure 14. Surface solar radiation anomalies, zenith transmittance, and maximum transmittance measured at individual stations.**

[Figure]

**Figure 15. Surface solar radiation anomalies of the 14 Japanese stations categorized into groups in alternative ways. In the sensitivity cases above, the Ishigakijima and Wajima stations, which is in the pristine group in the reference case, are switched to the polluted group one by one (cases I and III) and altogether (case II). See**
5    **the caption and legend for Figure 12.**

[Figure]

**Figure 16. Zenith transmittance of the 14 Japanese stations categorized into groups in alternative ways. See the caption for Figure 15.**

[Figure]

**Figure 17. Maximum transmittance of the 14 Japanese stations categorized into groups in alternative ways. See the caption for Figure 15.**

---

## Referee Report (RR1)

I think the paper could be published after a technical correction.

[Referee's comment] 3. Figs. 5, 6, and 7. The legends indicating the urbanization (e.g., residential area) are necessary.

[Our response] We are afraid that we do not understand what exactly this particular comment suggested. It may be unclear what the red and green rows mean when these figures are inspected alone. In the caption of Figure 5, we thus added the following sentence: "Rows for the eight polluted and six pristine observatories are indicated in red and green, respectively." We did not add this sentence to Figures 6 and 7 because the captions for Figures 6 and 7 refer to that for Figure 5.

[Response to authors]

The explanations for the map symbols showing the urban area are necessary.

I guess that the symbols, , and  show buildings or residential area.

---

## Author Response (AR2)

**NATIONAL INSTITUTE FOR ENVIRONMENTAL STUDIES**
CENTER FOR GLOBAL ENVIRONMENTAL RESEARCH

**DR. KATSUMASA TANAKA**
16-2 Onogawa, 305-8506 Tsukuba, JAPAN
Phone:+81-29-850-2493
tanaka.katsumasa@nies.go.jp
http://www.nies.go.jp

*Copernicus Gesellschaft mbH*
*Bahnhofsallee 1e*
*37081 Göttingen*
*Germany*
25 October 2016

Dear Prof. Toshihiko Takemura,

Thank you for accepting our paper "Is global dimming and brightening in Japan limited to urban areas?" by K. Tanaka, A. Ohmura, D. Folini, M. Wild, and N. Ohkawara for publication in *Atmospheric Chemistry and Physics (ACP)*.

Regarding the technical correction, we added the following text to the caption of Figure 5: "Hatched areas and solid dots are built-in areas and buildings/houses etc., respectively, indicating the area of urbanization." The captions for Figure 6 and 7 refer to that of Figure 5. With this change, we think that we made the technical correction suggested.

Yours sincerely,

Katsumasa Tanaka